# Translation and codon usage regulate Argonaute slicer activity to trigger small RNA biogenesis

Meetali Singh [1], Eric Cornes [1], Blaise Li [1,2], Piergiuseppe Quarato [1,3], Loan Bourdon [1], Florent Dingli [4], Damarys Loew [4], Simone Proccacia [1,5] & Germano Cecere [1✉]

In the *Caenorhabditis elegans* germline, thousands of mRNAs are concomitantly expressed with antisense 22G-RNAs, which are loaded into the Argonaute CSR-1. Despite their essential functions for animal fertility and embryonic development, how CSR-1 22G-RNAs are produced remains unknown. Here, we show that CSR-1 slicer activity is primarily involved in triggering the synthesis of small RNAs on the coding sequences of germline mRNAs and post-transcriptionally regulates a fraction of targets. CSR-1-cleaved mRNAs prime the RNA-dependent RNA polymerase, EGO-1, to synthesize 22G-RNAs in phase with translating ribosomes, in contrast to other 22G-RNAs mostly synthesized in germ granules. Moreover, codon optimality and efficient translation antagonize CSR-1 slicing and 22G-RNAs biogenesis. We propose that codon usage differences encoded into mRNA sequences might be a conserved strategy in eukaryotes to regulate small RNA biogenesis and Argonaute targeting.

[1] Mechanisms of Epigenetic Inheritance, Department of Developmental and Stem Cell Biology, Institut Pasteur, UMR3738, CNRS, Paris, France. [2] Hub de Bioinformatique et Biostatistique—Département Biologie Computationnelle, Institut Pasteur, Paris, France. [3] Sorbonne Université, Collège Doctoral, Paris, France. [4] Institut Curie, PSL Research University, Centre de Recherche, Laboratoire de Spectrométrie de Masse Protéomique, Paris, France. [5] Università di Trento, Trento TN, Italy. ✉email: germano.cecere@pasteur.fr

In animals, small RNAs expressed in the germline and transmitted to the embryo act as a defense mechanism to repress foreign RNAs such as viruses, transposons, and other repetitive elements (REs). These small RNAs are essential for fertility and genome integrity[1,2]. Their function is controlled by the conserved family of Argonaute proteins (AGOs), which loads the small RNAs and functions to repress complementary messenger RNA (mRNA) targets through their endonuclease activity or by recruiting other effector silencing proteins[3–6]. The *C. elegans* germline contains a complex small RNA regulatory network, with different classes of small RNAs, multiple AGO effectors, and diverse biogenesis pathways[7]. One of the most abundant classes of endogenous small RNAs in the germline is the 22G-RNAs, which are single-stranded antisense small RNAs produced by RNA-dependent RNA polymerase (RdRPs) as part of an amplification system to silence target transcripts (reviewed in [7]). The production of 22G-RNAs targeting REs is triggered by over 15,000 PIWI-interacting RNAs (piRNAs or 21U-RNAs) and loaded by Worm-specific Argonautes (WAGOs) to silence REs, including the nuclear Argonaute HRDE-1[8–12]. 22G-RNAs are also produced from most germline-expressed mRNAs by the RdRP EGO-1 and loaded into the Argonaute CSR-1[13,14]. In contrast to the 22G-RNAs antisense to REs, which can be triggered in response to piRNAs, the primary trigger for generating CSR-1 22G-RNAs and why many germline mRNAs become targeted by CSR-1 is still unknown (Supplementary Fig. 1).

Given that the *C. elegans* piRNAs can trigger their targets' silencing by imperfect complementarity, and therefore potentially target germline-expressed mRNAs[15–17], the targeting by CSR-1 22G-RNAs can function as an anti-silencing mechanism to protect germline mRNAs from piRNAs silencing[14,18,19]. The anti-silencing function of CSR-1 can occur in the nucleus or P granules. In the nucleus, CSR-1 has been shown to interact with chromatin in a 22G-RNA-dependent manner[14] where it can counteract piRNA-mediated silencing by antagonizing the binding of the nuclear Argonaute protein HRDE-1 to nascent germline transcripts[20,21]. In P granules, CSR-1 can scan the mRNAs exiting the nuclear pore and compete with piRNA targeting[15]. The anti-silencing function of CSR-1 was primarily established with single-copy transgenes[16,18,19]. However, germline mRNAs remain protected from piRNAs silencing even in the absence of CSR-1[17], and sequence-encoded features of germline mRNAs have also been proposed to prevent piRNA silencing[15,17]. To what extent endogenous germline-expressed genes are regulated by CSR-1 and piRNA pathways' antagonistic functions remain elusive (Supplementary Fig. 1).

In addition, CSR-1 has been proposed to regulate the expression of its germline targets directly. Transcriptomic analyses of CSR-1 loss of function alleles have shown that CSR-1 promotes the expression of its target genes in hermaphrodites and males[14,22,23]. On the other hand, of the Argonautes that load 22G-RNAs, only CSR-1 has demonstrated slicer activity on target mRNA in vitro[24], and worms expressing a CSR-1 catalytic mutant protein show upregulation of its germline target genes[25]. Thus, it remains unclear whether CSR-1 positively or negatively regulates the expression of its target mRNAs. This is because all these studies have been performed using different methodologies at different developmental stages using either CSR-1 mutants, hypomorphs, or CSR-1 KO rescued with transgenic CSR-1 catalytic mutant[14,22,23,25]. As a result, the gene expression changes observed in the different studies do not largely overlap (Supplementary Fig. 2a, b)[14,23,25]. Therefore, the gene regulatory functions of germline CSR-1 22G-RNAs remain incompletely understood (Supplementary Fig. 1).

Similarly, the biogenesis of CSR-1 22G-RNAs remains mysterious. Many germline Argonautes, including CSR-1 and PIWI,

and proteins involved in 22G-RNA biogenesis, including RdRPs, localize to germ granules[14,26]. These germ granules are thought to be the site for the biogenesis of all germline 22G-RNAs. Germ granules are organized in sub-compartments—M, Z, and P granules[27]. Disruption of M granule (also known as mutator foci), which participates in piRNA-dependent 22G-RNA production, has no apparent effect on CSR-1 22G-RNAs[28,29]. Moreover, the type of RNA template used by the EGO-1 RdRP to generate CSR-1 22G-RNAs remains mysterious. During exogenous RNAi, the addition of alternating non-templated uridine (U) and guanosine (G) ribonucleotides (polyUG) to the 3′ termini of cleaved mRNA targets by RDE-3 recruits RdRPs EGO-1 and RRF-1 to synthesize 22G-RNAs[30,31]. However, RDE-3 is not required to generate CSR-1 22G-RNAs[26,31]. Thus, the subcellular location and RNA substrate used to create 22G-RNAs is unknown.

In the current study, we elucidate CSR-1 catalytic activity-dependent and independent germline gene regulation and decipher the rules governing CSR-1 22G-RNA biogenesis. We demonstrate that the slicer activity of CSR-1 triggers the biogenesis of 22G-RNAs antisense to the coding sequence of germline mRNAs. We establish that CSR-1 22G-RNAs are synthesized on an actively translated mRNA template in the cytosol, independent of germ granules. Overall, this study establishes that translation and codon usage dictate CSR-1 slicer activity on a target mRNA to regulate small RNA biogenesis and functions.

## Results

**Defects in CSR-1 catalytic activity mainly impact 22G-RNA abundance.** Both *csr-1* catalytic mutant (*csr-1* ADH) and knockout (*csr-1* KO) worms show reduced fertility and 100% embryonic lethality[32]. However, their gene expression profiles are different (Supplementary Fig. 2a, b). We hypothesized that the global impact of CSR-1 mutations on gene expression might depend on the developmental context and might be biased by developmental defects[14,33]. Indeed, we observed differences during oogenesis in *csr-1* ADH and *csr-1* KO worms marked by a delayed onset of oocyte production and increased accumulation of oocytes in the germline in *csr-1* ADH at a more advanced age compared to wild-type (WT) (Supplementary Fig. 2c–f). To overcome this limitation, we developed a sorting strategy to obtain a synchronized population of WT and first-generation homozygotes for *csr-1* KO or *csr-1* ADH strains using COPAS biosorter, which allowed us to collect almost a pure population of M+/Z− mutants (Supplementary Fig. 2g). Using this strategy, we enriched for larval stage late L4 worms, characterized by a closed vulva and absence of oocytes and lacking the germline developmental abnormality (Supplementary Fig. 2h, i).

Next, to precisely evaluate the role of CSR-1, we measured small RNA accumulation (sRNA-seq), transcription (GRO-seq), mRNA stability (RNA-seq), and translation (Ribo-seq) in WT and mutant worms. In addition, to assess the direct effect of CSR-1 22G-RNAs on these processes, we sequenced the small RNAs bound to immunoprecipitated CSR-1 from similarly sorted late L4 worms to precisely identify the CSR-1 targets at the same developmental stage. We detected a total of 4803 genes with antisense 22G-RNAs loaded into CSR-1 (IP over input ≥ twofold enrichment and RPM ≥ 1 in each replicate of CSR-1 IP) (Supplementary Data 1). These mRNA targets are germline enriched and largely overlap with previously defined targets[14,22] with some variations based on developmental stages studied (Supplementary Fig. 3a, b). The *csr-1* ADH displayed a global loss of 22G-RNAs for the majority of CSR-1 targets (Fig. 1a, c). However, only 7.7% (*n* = 119) of CSR-1 targets with >2-fold reduction of 22G-RNAs (*n* = 1536) showed increased mRNA levels, and only one showed twofold downregulation (Fig. 1b),

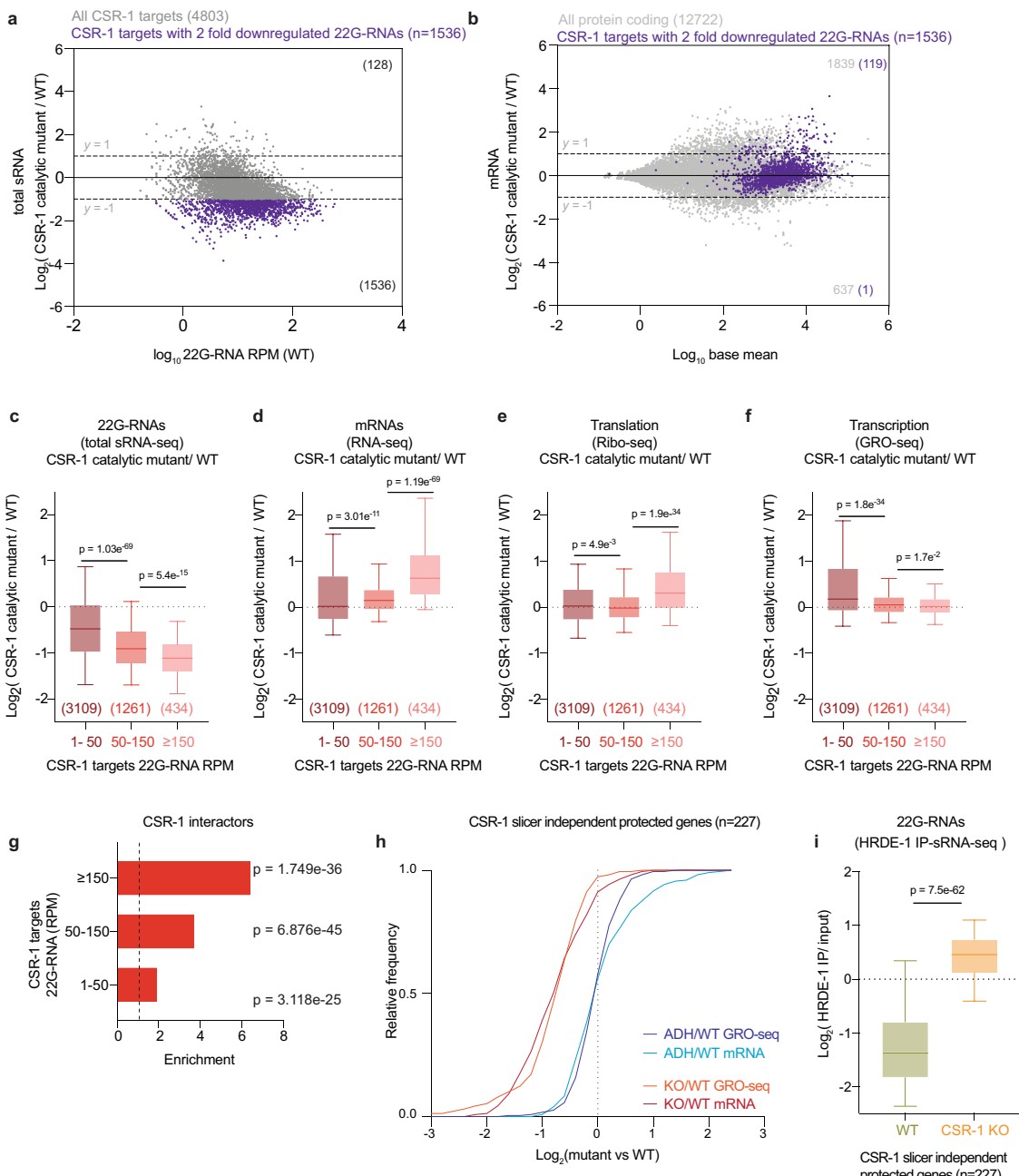

**Fig. 1 Defects in CSR-1 catalytic activity mainly impacts 22G-RNA abundance. a** MA-plot showing total 22G-RNA $\log_2$ fold-change for CSR-1 ADH (catalytic mutant) compared to WT. The number in parenthesis indicates the number of misregulated genes ≥2-fold. The average from two biological replicates is shown. **b** MA-plot showing mRNA $\log_2$ fold-change for CSR-1 ADH compared to WT. Genes with 22G-RNAs with twofold downregulation in CSR-1 ADH compared to WT are highlighted in purple. The average from two biological replicates is shown, with "base mean" computed using DESeq2[89]. The number in parenthesis indicates the number of misregulated genes ≥2-fold (gray dots—all protein-coding genes, purple- CSR-1 targets with twofold downregulated 22G-RNAs). **c–f** Box plots showing the $\log_2$ fold-change in CSR-1 ADH compared to WT strain for total 22G-RNAs (sRNA-seq) (2 biological replicates) (**c**); or mRNAs (RNA-seq) (2 biologically replicates) (**d**); mRNAs engaged in translation (Ribo-seq) (3 biological replicates) (**e**); and nascent RNAs (GRO-seq) (2 biologically replicates) (**f**), The distribution for the CSR-1 targets with 22G-RNA in CSR-1 IP with 1–50 RPM, 50–150 RPM, or ≥150 RPM is shown (gene list in Supplementary Data 1). **g** Enrichment of CSR-1 interactors in different CSR-1 targets categories based on 22G-RNA abundance. The dashed line at 1 indicates no enrichment. *P*-values were calculated by Exact hypergeometric probability using an automated tool available at http://nemates.org/MA/progs/overlap_stats.html. **h** Cumulative frequency distribution for CSR-1 slicer-independent protected targets (downregulated in CSR-1 KO compared to the CSR-1 ADH in GRO-seq, gene list in Supplementary Data 1). The comparison shows GRO-seq ($P = 1.6e^{-49}$) and RNA-seq ($P = 4.2e^{-37}$) for CSR-1 KO or CSR-1 ADH compared to WT. **i** Box plots showing the $\log_2$ fold-change of 22G-RNAs (sRNA-seq) in HRDE-1 IPs compared to input in WT, CSR-1 KO strains. Data is representative of two biological replicates. For all the box plots, the line indicates the median value, the box indicates the first and third quartiles, and the whiskers indicate the 5th and 95th percentiles, excluding outliers. Two-tailed *P*-values were calculated using Mann–Whitney–Wilcoxon tests. The sample size n (genes) is indicated in parentheses. Source data are provided as a Source Data file.

indicating that most mRNA targets are not destabilized by CSR slicer activity. We also detected some targets with upregulated levels of 22G-RNAs (Fig. 1a), which belong to spermatogenic genes and are being investigated in an independent study. We further divided CSR-1 targets into three bins based on 22G-RNA amounts loaded by CSR-1 in IP and analyzed gene expression changes and dependence on 22G-RNA levels. The increase in mRNA and translational levels of the targets in *csr-1* ADH correlated with 22G-RNA levels in CSR-1 IPs in a dose-dependent manner (Fig. 1d, e) in agreement with a previous report[25], but their transcription was unaffected (Fig. 1f). Therefore, our results support a previously developed model that CSR-1 slices a subset of mRNA targets having abundant 22G-RNAs at the post-transcriptional level[25,32]. Moreover, CSR-1 interactors identified by mass spectrometry (MS/MS) are enriched with CSR-1 targets that are post-transcriptionally regulated by CSR-1 (Fig. 1g and Supplementary Fig. 3c). Most of these targets are direct interactors and are not impacted by RNase treatment (Supplementary Fig. 3d–f). Thus, CSR-1 slicer activity negatively regulates the expression of its own interactors, including CSR-1, suggesting a negative feedback loop.

Overall, these results suggest that the main role of CSR-1 catalytic activity is to control the accumulation of CSR-1 interacting 22G-RNAs. In addition, CSR-1 post-transcriptionally regulates a small fraction of CSR-1 targets that have highly abundant 22G-RNAs.

**CSR-1 protects a subset of oogenic enriched targets from piRNA-mediated transcriptional silencing**. Similar to *csr-1* ADH worms, *csr-1* KO worms displayed a loss of 22G-RNAs as well as an upregulation of a subset of target mRNAs characterized by a high abundance of 22G-RNAs (Supplementary Fig. 4a–d). However, the level of upregulation of CSR-1 target mRNAs was significantly lower in the *csr-1* KO compared to the *csr-1* ADH, possibly due to decreased transcription (Supplementary Fig. 4e). Indeed, we found that a subset of target genes displayed down-regulated transcription and reduced mRNA levels in the KO compared to *csr-1* ADH. These were downregulated in KO compared to WT but were unaffected in the *csr-1* ADH (Fig. 1h). The majority of these genes (53%) were enriched for oogenic mRNAs (see Supplementary Data 1 for gene list) (Supplementary Fig. 4f), and there was no clear correlation with the abundance of 22G-RNAs loaded by CSR-1 for these targets. Given that CSR-1 is proposed to protect germline transcripts from piRNA-mediated silencing, we hypothesized that in the *csr-1* KO, piRNAs can trigger the loading of 22G-RNAs into the nuclear Argonaute HRDE-1 resulting in the reduced transcription of this subset of CSR-1 targets. We observed an increased number of CSR-1 targets with their 22G-RNAs being loaded by HRDE-1 in *csr-1* KO compared to WT (Supplementary Fig. 4g). Indeed, we noticed HRDE-1 loads increased levels of 22G-RNAs from transcriptionally downregulated CSR-1 targets in the *csr-1* KO (Fig. 1i). These experiments provide evidence that endogenous genes can be targeted by HRDE-1 in the absence of CSR-1, supporting its anti-silencing role. We further show that CSR-1 sliced targets and CSR-1 protected targets are mutually exclusive (Supplementary Fig. 4h), highlighting a slicer-dependent regulation of gene expression and slicer-independent role in protecting a subset of oogenic targets from piRNA-mediated HRDE-1 transcriptional silencing.

**CSR-1 catalytic activity is required for biogenesis of 22G-RNAs antisense to the coding sequence of target mRNAs**. The global reduction of CSR-1-bound 22G-RNAs observed in CSR-1 mutants, including CSR-1 sliced as well as CSR-1 protected

targets (Supplementary Fig. 4i), suggests that CSR-1 catalytic activity is required for 22G-RNA loading or biogenesis. Despite the reduction in total 22G-RNAs in the *csr-1* ADH strain, an enrichment of 22G-RNAs in IP over input was observed for CSR-1 ADH protein (Supplementary Fig. 5a, b), suggesting that catalytic inactive CSR-1 (CSR-1 ADH) can still bind the 22G-RNAs produced in the mutant. In fact, CSR-1 ADH showed enhanced binding efficiency compared to WT CSR-1, suggesting that either the loading of 22G-RNA is more efficient in CSR-1 ADH or the catalytic mutant protein stabilizes its interacting 22G-RNAs.

We then investigated the distribution of CSR-1-bound 22G-RNAs along the target-gene bodies. We found that the reduction in 22G-RNAs in *csr-1* ADH and KO primarily occurred antisense to the coding sequence (CDS) of CSR-1 targets, whereas 22G-RNAs derived from the 3′-untranslated region (3′UTR) were largely unaffected (Fig. 2a–e and Supplementary Fig. 5c). These results indicate that the RdRP fails to synthesize 22G-RNAs on the CDS in the absence of catalytic activity.

The RdRP EGO-1 has been proposed to exclusively synthesize CSR-1-bound 22G-RNAs[13,14,26]. We confirmed these results by using an *ego-1* knockout (KO) and sequenced 22G-RNAs, which were depleted both at CDS and 3′UTR (Fig. 2d, f). To understand whether the small RNAs produced on the 3′UTR in the absence of CSR-1 protein or its catalytic activity are also synthesized by EGO-1, we efficiently depleted CSR-1 using an auxin-induced degradation system, combined with *ego-1* knockdown by RNAi (Supplementary Fig. 5d–f). First, we confirmed that CSR-1 22G-RNAs were depleted on CDS and enriched on 3′UTR upon auxin-induced CSR-1 depletion (Fig. 2g, h). Next, we observed reduced 22G-RNAs from both CDS as well as 3′UTR upon *ego-1* knockdown by RNAi (Fig. 2g, i and Supplementary Fig. 5g, h), implying that EGO-1 may be exclusively responsible for the synthesis of the CSR-1 22G-RNAs in both WT and the *csr-1* mutants. However, the catalytic activity of CSR-1 is required to efficiently generate EGO-1-dependent 22G-RNAs along the coding sequences of target mRNAs. To understand if another class of endogenous small RNAs, the 26G-RNA[34,35], may be priming the EGO-1 recruitment at the 3′UTR, we combined *ego-1* RNAi with mutant of RdRP, *rrf-3*−/−, which is responsible for 26G-RNA production[34,35]. However, we did not observe any contribution of RRF-3 produced 26G-RNAs in the biogenesis of CSR-1 22G-RNAs and EGO-1 priming on 3′UTR of CSR-1 targets (Supplementary Fig. 6a–c). Also, we did not observe any compositional bias for 22-nt small RNAs derived from CDS and 3′UTR (Supplementary Fig. 6d, e). CSR-1 -associated 22G-RNAs can also be poly uridylated (U)[36]. Thus, we investigated whether there was any difference in the levels of CSR-1 22G-RNAs poly (U) in both *csr-1* ADH and *csr-1* KO compared to WT. Our analysis showed a loss of CSR-1-associated 22G-RNAs poly (U) both at CDS and 3′UTR (Supplementary Fig. 6f–h), similar to what we have observed for total CSR-1 22G-RNAs. The CSR-1 22G-RNAs poly(U) were also globally reduced in *ego-1* KO (Supplementary Fig. 6f, g), suggesting poly(U) addition happens post-22G-RNA biogenesis[36]. Thus, how EGO-1 is recruited at 3′UTR remains to be investigated.

Finally, we tested whether the restored expression of CSR-1 is sufficient to generate EGO-1-dependent 22G-RNAs on the gene body. For this purpose, we depleted CSR-1 by auxin-induced degradation for 38 h after hatching (0 h recovery) and then reintroduced CSR-1 by recovering expression for 5 and 10 h (Supplementary Fig. 6i). As expected, the depletion of CSR-1 caused a loss of 22G-RNA accumulation on the CDS (Fig. 2j and Supplementary Fig. 6i—see 0 h recovery). However, upon reintroduction of CSR-1 expression (5 and 10 h recovery), we observed a steady increase of 22G-RNAs, mainly on the CDS (Fig. 2j, k). The lack of complete recovery of 22G-RNAs could be

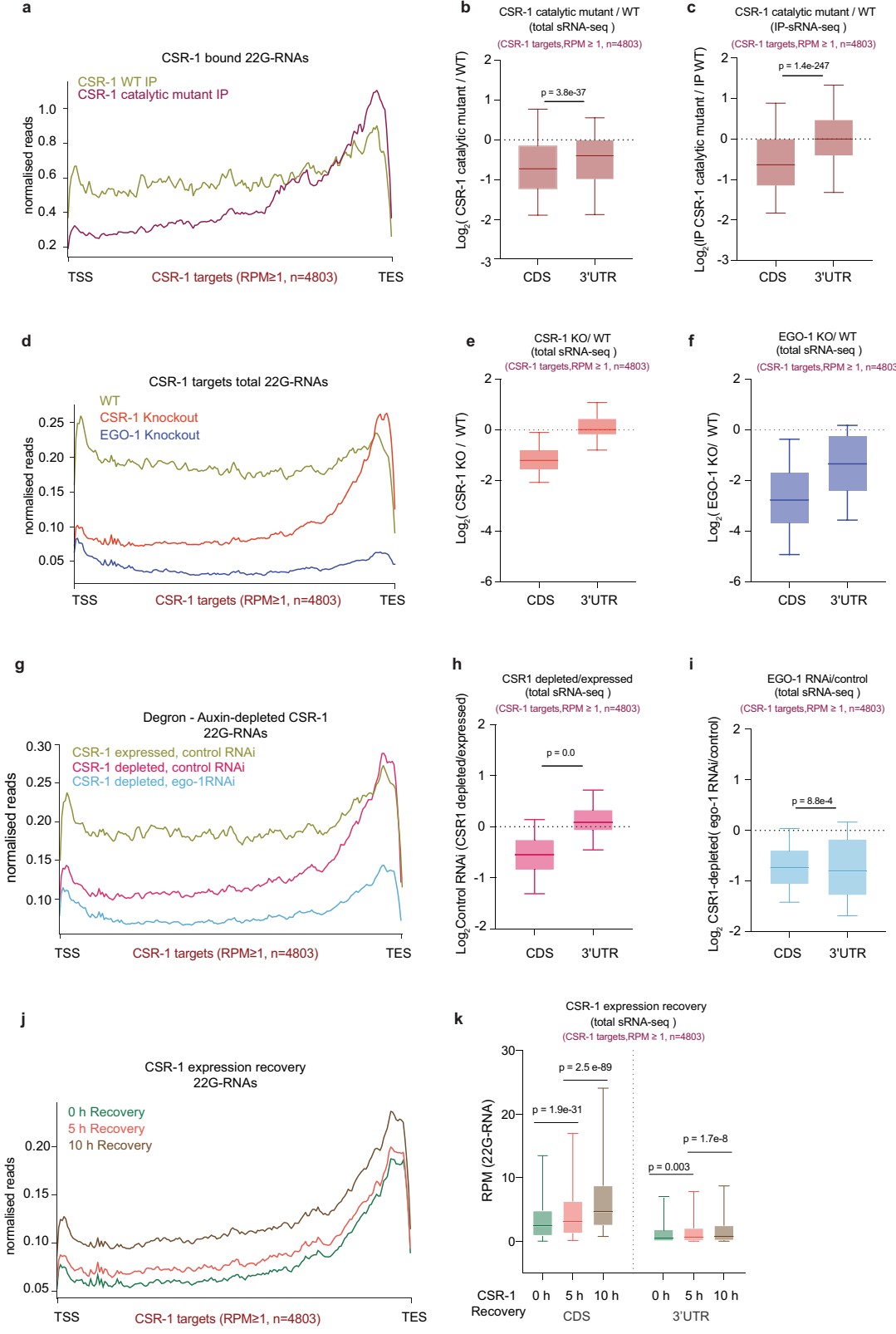

due to the accumulation of germline defects as a result of CSR-1 depletion during the initial period of germline development.

Overall, these data demonstrate that EGO-1 can be recruited on the 3′UTR of target mRNAs and initiate the production of 22G-RNAs. However, CSR-1-mediated slicing of mRNAs is required to template the production of small RNAs on the gene body.

**Biogenesis of CSR-1 22G-RNAs and the regulation of their targets occurs in the cytosol.** PIWI and RNAi biogenesis factors are known to localize in perinuclear condensates, called germ granules, and these germ granules have been proposed to be the site for biogenesis of 22G-RNAs[28,37–39]. Germ granules have been shown to be organized in different compartments, namely P, M (also known as mutator foci), and Z granules[27]. CSR-1 and

**Fig. 2 CSR-1 catalytic activity is required for the biogenesis of 22G-RNAs antisense to the coding sequence. a, d, g, j** Metaprofile analysis showing the distribution of normalized 22G-RNA (sRNA-seq) reads (RPM) along all CSR-1 targets (≥1 RPM, n = 4803) in WT CSR-1 or CSR-1 ADH immunoprecipitation (IP) (**a**); in WT, *csr-1* KO and *ego-1* KO (**d**); upon *ego-1* RNAi and Control RNAi treated in Auxin-depleted CSR-1 degron background (CSR-1 depleted) and degron control (CSR-1 expressed) (**g**); after depletion of CSR-1, in the CSR-1 degron strain for 38 h by growing on auxin containing plates and recovery of CSR-1 expression by transferring on plates without auxin for 0, 5, and 10 h (**j**), TSS indicates the transcriptional start site, TES indicates the transcriptional termination site. An average of two biological replicates is shown. **b, c** Box plots showing the log$_2$ fold-change of the amount of total 22G-RNA generated from CDS and 3′UTR of all CSR-1 targets in CSR-1 ADH compared to WT (**b**); 22G-RNA generated from CDS and 3′UTR of all CSR-1 targets bound in CSR-1 ADH IP compared to WT CSR-1 IP (**c**). **e, f** Box-plot showing the log$_2$ fold-change in the amount of 22G-RNA generated from CDS and 3′UTR of all CSR-1 targets in *csr-1* KO compared to WT (**e**) and in *ego-1* KO compared to WT (**f**). **h, i** Box-plot showing the log$_2$ fold-change in the amount of 22G-RNA generated from CDS and 3′UTR of all CSR-1 targets in Auxin-depleted CSR-1 compared to non-depleted CSR-1 degron control in control RNAi background (**h**); for *ego-1* RNAi compared to control RNAi treated in Auxin-depleted CSR-1 degron background (**i**). **k** Box-plot representing the data in **j** showing the RPM of 22G-RNAs generated from CDS and 3′UTR of CSR-1 targets (22G-RNA ≥ 1 RPM) for CSR-1 expression recovered for 0, 5, or 10 h. For all the box plots, the line indicates the median value, the box indicates the first and third quartiles, and the whiskers indicate the 5th and 95th percentiles, excluding outliers. Two-tailed *P*-values were calculated using Mann–Whitney–Wilcoxon tests. For all the experiments, the sample size *n* (genes) is indicated in parentheses, which include all CSR-1 targets. For all experiments, data is representative of two biological replicates. Source data are provided as a Source Data file.

EGO-1 localize in both cytosol and the P granules[14], suggesting that the biogenesis of CSR-1 22G-RNAs might also occur in these organelles. To test this possibility, we used RNAi to simultaneously deplete four core components of P granules (*pgl-1, pgl-3, glh-1*, and *glh-4*)[40], (Supplementary Fig. 7a). This treatment was sufficient to disrupt not only P granules but also mutator foci and Z granules as observed by imaging of their respective components PGL-1 and DEPS-1 (P granule), MUT-16 (mutator foci), and ZNFX-1 (Z granule) (Fig. 3a). Mutator foci were previously not shown to be disrupted by RNAi against either single or two components of P granule[28]. However, RNAi against four P granule components disrupts mutator foci also (Fig. 3a). The treatment also depleted the majority of CSR-1 localization in P granules (Fig. 3b). However, the cytosolic localization of CSR-1 remained unaffected (Fig. 3b). We still observed a residual granular localization of CSR-1, which we attribute to a lack of 100% knockdown during RNAi treatment. In fact, remaining CSR-1 localized with residual DEPS-1 (a component of P granule) upon P granule RNAi (Fig. 3b). Z granule mutant (*znfx-1*) or mutator foci mutant (*mut-16*) did not affect CSR-1 localization to P granule (Supplementary Fig. S7b).

Next, we evaluated the effects of loss of germ granules on 22G-RNA biogenesis. Though piRNA-dependent 22G-RNAs were globally depleted upon P granule RNAi treatment (Fig. 3c), CSR-1 22G-RNAs were unaffected upon P granule RNAi treatment, despite the loss of perinuclear CSR-1 P granule localization (Fig. 3b, c). Furthermore, CSR-1 targets were not upregulated upon P granule RNAi (RNA-seq data from [41]) (Fig. 3d and Supplementary Fig. 7a). Though a synthesis of CSR-1 22G-RNAs in P granules cannot be completely ruled out, these results highlight that majority of CSR-1 22G-RNA biogenesis occurs in the cytosol, and P granules are dispensable.

**Translating mRNAs serve as the template for 22G-RNA biogenesis.** Our data so far suggest that majority of CSR-1 22G-RNAs might be generated in the cytosol. Consistent with CSR-1 localization in the cytosol and P granules, we identified ribosomal and ribosomal-associated proteins, which are enriched in the cytosol, and germ granule components in our immunoprecipitation-mass spectrometry (IP-MS/MS) as direct CSR-1 interactors (Fig. 4a). Ribosomal interactors of CSR-1 were not lost upon RNase treatment, contrary to ribosomal interactors of PIWI, which are lost on RNase treatment (Supplementary Fig. 7c), suggesting that CSR-1 directly interacts with ribosomal proteins. Moreover, CSR-1 ADH showed reduced co-purification of ribosomal proteins and increased co-purification of P granule components, compared to CSR-1 WT (Fig. 4b). The catalytic mutation leads to an enriched

localization of CSR-1 ADH within P granules, as can be seen with co-localization with GLH-1 (a component of P granule) in enlarged granules and this increased expression is consistent with the observation that CSR-1 self-regulates its expression (Fig. 4c).

Based on these data, we hypothesized that 22G-RNAs are synthesized in the cytosol, using translating mRNAs as templates. To test this hypothesis, we performed polysome profile and immunoblot for CSR-1 and EGO-1, which were both enriched in polysome fractions, suggesting they interact with translating mRNAs (Fig. 4d). In contrast, PIWI and PGL-1 (a P granule component) were not enriched in the polysome fractions, further supporting the synthesis of PIWI-dependent 22G-RNAs in P granules, which are devoid of mRNAs engaged in translation[42].

We then mapped the distance between the start of the 29-nucleotide Ribosomal Protected Fragments (RPF)[43] and the 5′-end of CSR-1 22G-RNAs (Supplementary Fig. 7d). We observed the characteristic three-nucleotide (3-nt) periodicity pattern typical of ribosomal footprints (Fig. 4e), indicating that the synthesis of CSR-1 22G-RNAs occurs on mRNA templates engaged in translation in phase with the ribosome. In contrast, the HRDE-1 loaded 22G-RNAs of P granule-dependent piRNA targets (Supplementary Data 1) did not show phasing with ribosomes as observed due to a lack of three-nucleotide periodicity and were randomly distributed (Fig. 4e and Supplementary Fig. 7d), in agreement with the fact that P granules are devoid of translating mRNAs[42,44] and PIWI is not enriched in polysome fractions. Altogether these results suggest that CSR-1 cleaves actively translating mRNAs, which become the template for EGO-1-mediated synthesis of 22G-RNAs on the coding sequence of mRNA targets.

**mRNA translation antagonizes CSR-1 22G-RNA biogenesis.** EGO-1-mediated synthesis of CSR-1 22G-RNAs does not occur on every germline mRNA at similar levels, and we found that the levels of 22G-RNA are independent of the levels of the mRNA template (Supplementary Fig. 8a). Given our observations that, actively translating mRNAs serve as the template for CSR-1 22G-RNAs, we hypothesized that the translation efficiency (TE) of germline mRNAs impacts CSR-1 22G-RNA biogenesis. To test this hypothesis, we calculated the TE of CSR-1 targets using the Ribo-seq and RNA-seq data from WT worms at the late L4 stage. We observed that levels of CSR-1-associated 22G-RNAs produced from a target mRNA were inversely correlated with their TE (Fig. 5a), suggesting that translation antagonizes the biogenesis of CSR-1 22G-RNAs.

Codon usage and the availability of the tRNA pool influence TE[45,46]. Therefore, we investigated whether these mechanisms affect the biogenesis of CSR-1 22G-RNAs. We determined

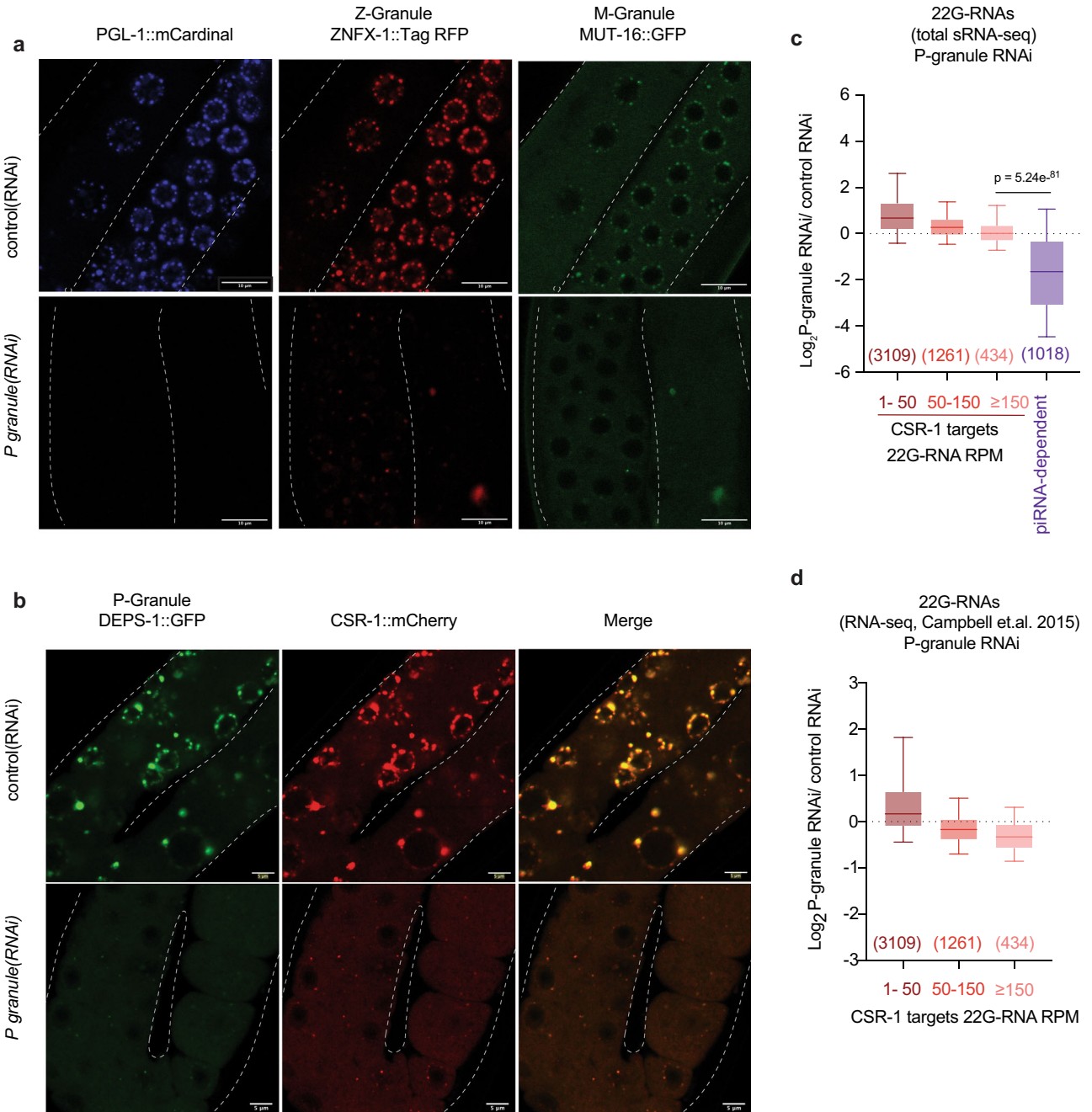

**Fig. 3 Biogenesis of CSR-1 22G-RNAs and the regulation of their targets do not require germ granule localization. a** Live-fluorescent images of animals expressing P granule marker PGL-1::mCardinal (P granule), ZNFX-1::TagRFP (Z granule), and MUT-16::GFP (Mutator foci) treated with either control RNAi or P granule RNAi (*pgl-1, pgl-3, glh-1,* and *glh-4*). All three granule types are depleted. Scale bars represent 10 μm. **b** Live-fluorescent image of animals expressing P granule marker DEPS-1::GFP (P granule) and mCherry::CSR-1 treated with either control RNAi or P granule RNAi (*pgl-1, pgl-3, glh-1,* and *glh-4*). Scale bars represent 5 μm. CSR-1 is localized in both cytosol and P granule, and upon RNAi treatment, P granule localization of CSR-1 is lost while maintaining cytosolic localization. At least ten individual germlines were imaged for each condition in **a**, **b**. **c** Box plots showing the log$_2$ fold-change of total 22G-RNA (sRNA-seq) upon P granule RNAi compared to control RNAi. The distribution for the 22G-RNA in CSR-1 IP for CSR-1 targets with 1–50 RPM, 50–150 RPM, or ≥150 RPM (gene list in Supplementary Data 1) and piRNA-dependent 22G-RNA target genes[66] are shown. Data is average of two biological replicates. **d** Box plots showing the log$_2$ fold-change of mRNA expression (RNA-seq from Campbell et. al.[41]) upon P granule RNAi compared to control RNAi. The distribution for the 22G-RNA in CSR-1 IP for CSR-1 targets with 1–50 RPM, 50–150 RPM, or ≥150 RPM. For both the box plots, the line indicates the median value, the box indicates the first and third quartiles, and the whiskers indicate the 5th and 95th percentiles, excluding outliers. Two-tailed *P*-values were calculated using Mann–Whitney–Wilcoxon tests. The sample size *n* (genes) is indicated in parentheses. Source data are provided as a Source Data file.

optimal and non-optimal codons using our experimental data from Late L4-staged worms. First, we calculated the normalized average relative synonymous codon usage (RSCU) for genes for different categories of high or low TE (Fig. 5b). Codons showing enrichment in genes with high TE (log$_2$TE ≥ 3) were considered optimal codons, and the ones under-represented were considered non-optimal codons (Fig. 5b). We confirmed that our classification of optimal/non-optimal codons correlated with tRNA copy

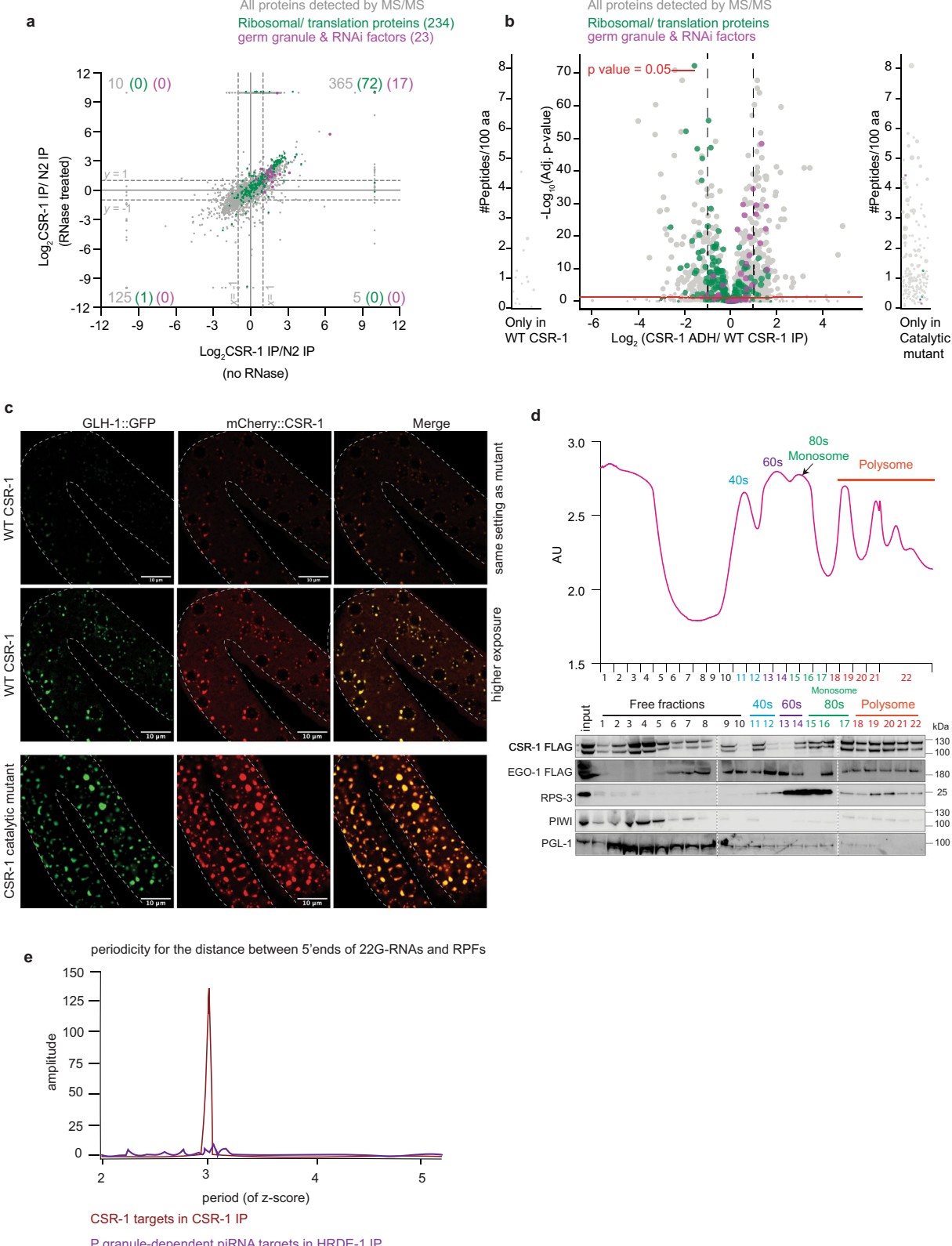

number (Fig. 5c and Supplementary Fig. 8b, d) and tRNA expression in the late L4 worm population (44 h) as measured by GRO-seq (Fig. 5d and Supplementary Fig. 8c, e). We noticed that for codons with no tRNA cognates and requiring tRNA binding by wobble pairing, all optimal codons end with C and non-optimal with U. Translation elongation is lower for those ending with a U[47].

We then evaluated the codon usage of CSR-1 targets by comparing their normalized average RSCU to highly translated mRNAs. We found that non-optimal codons were enriched, and optimal codons were depleted in CSR-1 targets, suggesting that this might be an encoded feature of mRNA targets influencing the priming of 22G-RNA synthesis (Fig. 5e). Non-optimal codons are known to promote ribosome stalling[48–50]. To map differences in

**Fig. 4 CSR-1 22G-RNAs are synthesized concomitantly with mRNA translation. a** Scatter plot comparing the log$_2$ fold-changes in CSR-1 interactors (IP-MS/MS) to control IPs performed in WT strain in the absence of RNase treatment (x-axis) to the IPs performed after RNase treatment (Supplementary Data 2). Ribosomal proteins and translation regulators are highlighted in green, and germ granule proteins, including RNAi factors, are highlighted in magenta. Number in gray refers to all interactors with log$_2$ fold-change of ≥1 and P-value ≤ 0.05 for each quadrant. The number in parenthesis is for ribosomal and translation-associated proteins enriched and granule and RNAi factors. n = 4 biological replicates. **b** Volcano plot showing log$_2$ fold-change in enrichment values and corresponding significance levels for proteins co-purifying with CSR-1 ADH compared to WT CSR-1 (Supplementary Data 3). Ribosomal proteins and translation regulators are highlighted in green. Germ granule proteins, including RNAi factors, are highlighted in magenta. The size of the dots is proportional to the number of peptides used for the quantification. The linear model was used to compute the protein quantification ratio, and the red horizontal line indicates the two-tailed P-value = 0.05. n = 4 biological replicates**. c** Live-fluorescent images showing localization and expression of GLH-1::GFP (P granule marker) and WT mCherry::CSR-1 or catalytic mutant mCherry::CSR-1 ADH. In WT, CSR-1 is localized to the cytosol and P granule. In CSR-1 catalytic mutant, CSR-1 is predominantly localized in enlarged P granules (Brightness of WT strain enhanced in middle panel for better visualization as the expression level of the mutant protein is higher than WT). At least five individual germlines were imaged for each strain. **d** Representative polysome profile indicating elution fractions with sub-monosomal, monosomal and polysomal complexes. Immunoblot for FLAG::CSR-1, FLAG::EGO-1 with anti-FLAG antibody and RPS-3, PIWI, and PGL-1 with their respective antibodies in sub-monosomal, monosomal, and polysomal fractions. The blots have been reproduced. **e** Periodogram based on Fourier transform for read-density around RPF 5′ start position showing periodicity of CSR-1 22G-RNAs phasing with RPFs. P granule-dependent piRNA targets in HRDE-1 IP were used as control. Data is representative of two biological replicates. Source data are provided as a Source Data file.

22G-RNA biogenesis on sequences with optimal or non-optimal codons, we divided RPFs into two categories based on the presence of either an optimal or non-optimal codon at the A and P sites of the ribosome. We did not observe any specific bias at the last position of RPF (Supplementary Fig. 8f). We then mapped the distance between 5′ of 22G-RNAs and RPFs and observed a peak for the 5′-end of 22G-RNAs downstream of RPF (29th position) when the A and P sites of the ribosomes are occupied by a non-optimal codon contrary to when optimal codons are present on A and P sites, which show no bias (Fig. 5f). This result suggests that the 22G-RNA production is preferentially initiated downstream of ribosomes especially occupying non-optimal codons that are difficult to translate, by CSR-1-mediated slicing and recruitment of EGO-1.

Altogether, these observations suggest that translation and ribosome position dictate the production of CSR-1 22G-RNAs.

**Increasing the translation efficiency of a CSR-1 target impairs CSR-1 22G-RNA biogenesis and function.** To determine whether non-optimal codons directly affect TE and CSR-1 22G-RNA biogenesis, we altered the coding potential of a CSR-1 target. We examined *klp-7*, which has the second-highest abundance of 22G-RNAs loaded by CSR-1 and is post-transcriptionally regulated by CSR-1. KLP-7 is a kinesin-13 microtubule depolymerase and is required for spindle organization and chromosome segregation[51]. Overexpression of KLP-7 in the *csr-1* mutant has been shown to cause microtubule assembly defects[25]. *klp-7* showed enrichment of non-optimal codons and depletion of optimal codons similarly to other CSR-1 targets (Supplementary Fig. 9a). We optimized the codon usage in *klp-7* by incorporating exclusively synonymous optimal codons (Supplementary Fig. 9a). We used CRISPR-Cas9 to replace endogenous *klp-7* isoform b with the modified *klp-7* codon-optimized (*klp-7*_co) to avoid disrupting potential UTR-mediated regulation.

To ascertain whether codon optimization of *klp-7* affected the TE, we performed RNA-seq and Ribo-seq from synchronized and sorted late L4 population (44 h). Indeed, we detected a twofold increase in the TE of *klp-7* mRNA in the *klp-7*_co strain compared to WT (Fig. 6a). The TE of other CSR-1 targets remained unaffected in the *klp-7*_co strain, indicating that the effects observed are specific to *klp-7* mRNA (Fig. 6b). In addition, KLP-7 protein levels were increased in two independent lines of *klp-7*_co compared to WT, consistent with increased translation (Supplementary Fig. 9b). We then measured the level of 22G-RNAs antisense to *klp-7* mRNA in the *klp-7*_co strain compared to WT, and we observed a 1.4-fold decrease in 22G-RNAs (Fig. 6a). The

levels of 22G-RNAs for other CSR-1 targets remained unaffected (Fig. 6b). Further, the significant decrease in 22G-RNAs on the optimized *klp-7*_co allele was observed in exons 3–6 and was accompanied by an increase in Ribo-seq peak height at those positions (Fig. 6c, d). The *klp-7*_co strain also showed increased *klp-7* mRNA level compared to WT (Fig. 6a), and we confirmed this result by quantitative reverse transcription PCR (RT-qPCR) (Supplementary Fig. 9c). These results suggest that CSR-1 targeting, and regulation are impaired on *klp-7*_co mRNA. To validate this, we performed *csr-1* RNAi and showed increased *klp-7* mRNA levels in the WT strain but not in the *klp-7*_co strain (Fig. 6e), suggesting that CSR-1 slicer activity is reduced at *klp-7*_co mRNA. The increased levels of *klp-7* mRNA correlated with a reduction in brood size (Supplementary Fig. 9d) and higher embryonic lethality at 25 °C in *klp-7*_co strain compared to WT (Supplementary Fig. 9e), indicating the physiological relevance of *klp-7* mRNA targeting by CSR-1. Finally, to rule out any difference in the production of either 22G-RNAs or mRNA levels due to possible developmental defects between *klp-7*_co and WT strain, we generated a heterozygote strain of *klp-7*_co with a fluorescent GFP marker on the balancer chromosome. We sorted heterozygote GFP-positive worms with one copy of modified *klp-7*_co and one copy of WT *klp-7* each and performed RNA-seq and sRNA-seq. We observed similar results with a 1.8-fold increase in mRNA levels for *klp-7*_co compared to the WT *klp-7* copy and a 1.25-fold decrease in 22G-RNA levels (Supplementary Fig. 9f, g). These results demonstrate that CSR-1 22G-RNA biogenesis and activity are reduced on mRNA templates with optimized codons and increased translation.

Altogether, these results suggest efficiently translating ribosomes block the access of CSR-1 to the mRNA template and thereby hamper 22G-RNA production and, in turn, affect gene regulation by CSR-1 during germline development.

## Discussion

In this study, we have determined the rules governing germline mRNA targeting by CSR-1 and addressed the long-standing paradox of CSR-1 function as an anti-silencer or a slicer. We show a significant fraction of the slicing activity of CSR-1 is directed towards the production of 22G-RNAs antisense the coding sequence of its targets. We further dissected the mechanism of CSR-1 22G-RNA production. We demonstrated that the majority of the synthesis of 22G-RNAs occurs in the cytosol on translating mRNA templates with a low translation efficiency. CSR-1 slices the target mRNA occupied by ribosomes and initiates 22G-RNA biogenesis by priming RdRP EGO-1

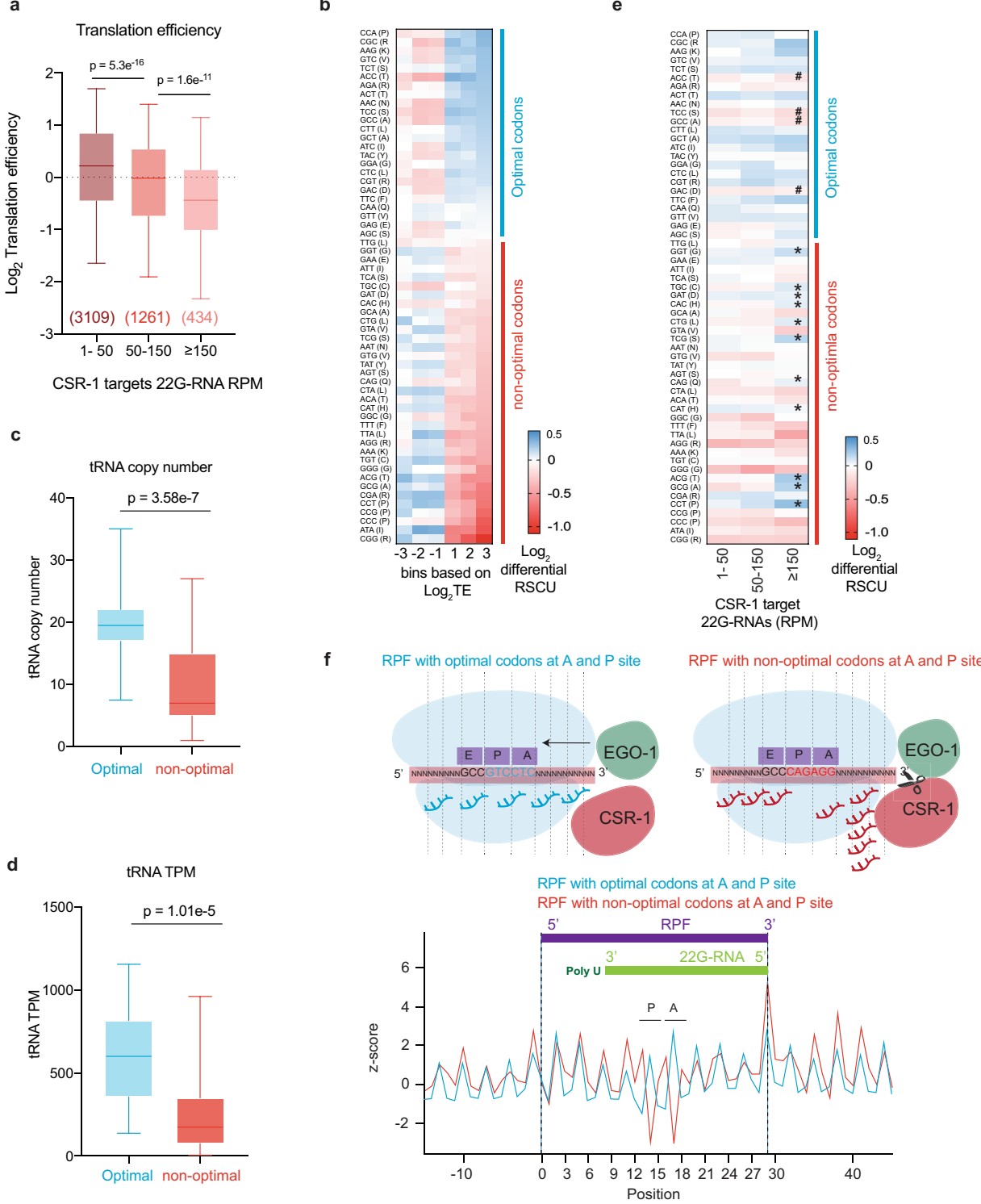

activity. Finally, we have determined how CSR-1 can preferentially target some germline mRNAs. We discovered that incorporating or avoiding non-optimal codons is a strategy adopted by germline mRNAs to be differentially regulated by CSR-1 22G-RNAs (Fig. 7). Overall, this study highlights the codon dependence and translational efficiency of mRNAs in the germline for the regulation of CSR-1 22G-RNAs biogenesis and, in turn, gene expression of the targets, which could have a significant bearing on germline gene regulation not just in worms but across species.

**CSR-1 function as slicer and anti-silencer**. We demonstrate that CSR-1 slicing activity regulates a fraction of targets, with a high abundance of 22G-RNA bound by CSR-1, post-transcriptionally in the germline, supporting the previous observation[25]. We further show that CSR-1 targets are enriched in CSR-1 direct interactors, including genes belonging to the CSR-1 pathway and CSR-1 itself. The upregulation of CSR-1 targets, therefore, may indirectly cause previously observed phenotypes, including chromatin defects[14,25,52]. Another recent study demonstrates that CSR-1 slicing activity is responsible for the decay of a larger

**Fig. 5 Highly translated mRNAs and optimal codons negatively correlate with CSR-1 22G-RNA abundance. a** Translation efficiency $\log_2$ (RPF TPM/ mRNA TPM) for CSR-1 targets in WT strain. The distribution for the 22G-RNA in CSR-1 IP for CSR-1 targets with 1–50 RPM, 50–150 RPM, or ≥150 RPM. The sample size $n$ (genes) is indicated in parentheses. The line indicates the median value, the box indicates the first and third quartiles, and the whiskers indicate the 5th and 95th percentiles, excluding outliers. Two-tailed $P$- values were calculated using Mann–Whitney–Wilcoxon tests. The sample size $n$ (genes) is indicated in parentheses. Data is an average of two biological replicates. **b** Heat map showing $\log_2$ fold-change in Relative Synonymous Codon Usage (RSCU) for all protein-coding genes categorized by increasing translational efficiency compared to genes showing neutral translational efficiency of 1, as explained in methods. Codons are arranged in order of decreasing RSCU (top to bottom) in the category of $\log_2$ TE ≥ 3. The blue line highlights optimal codons in genes with high TE, and the red line highlights non-optimal codons. **c, d** Box-plot showing the copy numbers for tRNAs for optimal or non-optimal codons (**c**), and the TPMs for tRNAs from the GRO-seq dataset for WT strain at the late l4 larval stage (44 h) for optimal or non-optimal codons (**d**). The line indicates the median value, the box indicates the first and third quartiles, and the whiskers indicate the 5th and 95th percentiles, excluding outliers. Two-tailed $P$-values were calculated using Mann–Whitney–Wilcoxon tests. Two biological replicates. For codons with missing cognate tRNA, values have been adjusted by considering tRNA copy numbers and TPMs for tRNA recognizing these codons by wobble base pairing[90]. (see Supplementary Fig. 8b, c for non-adjusted values). **e** Heatmap similar to **b** showing $\log_2$ fold-change in relative synonymous codon usage for all CSR-1 targets (1–50, 50–150, and ≥150 RPM of 22G-RNA) compared to genes showing neutral translational efficiency as explained in methods. "*" marks over-used non-optimal codons by CSR-1 targets and "#" marks under-used optimal codons. **f** Plot showing the $z$-score for the read-density for the of 5′ terminus of 22G-RNAs for CSR-1 targets relative to the start of 29-nt long Ribosomal protected fragments (RPF) with either optimal or non-optimal codons at their P and A site. The scheme shows possible initiation of 22G-RNA biogenesis after a slicing event downstream of RPF with non-optimal codons at the A and P sites. Source data are provided as a Source Data file.

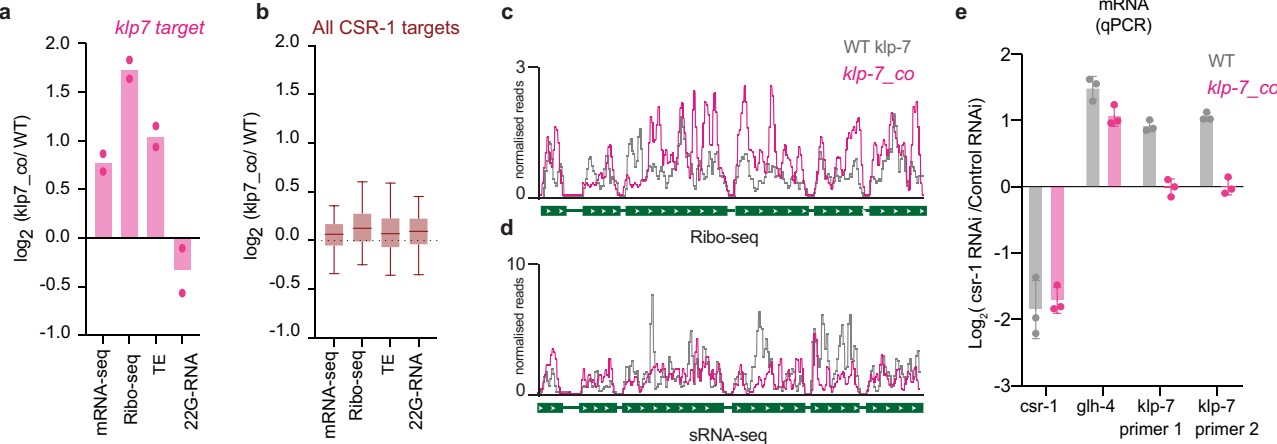

**Fig. 6 Increasing optimal codon usage of CSR-1 target decreases small-RNA production. a, b** Plot showing $\log_2$ fold-change for normalized reads for mRNAs from RNA-seq, RPF from Ribo-seq, and 22G-RNAs from sRNA-seq and differential Translational efficiency for *klp-7* (top CSR-1 target) (**a**) and all CSR-1 targets (RPM ≥ 1, $n = 4803$) (**b**) for the strain with codon-optimized *klp-7* (*klp7_co*) compared to WT strain. Two biological replicates are shown with their mean (**a**). For **b**, the line indicates the median value, the box indicates the first and third quartiles, and the whiskers indicate the 5th and 95th percentiles, excluding outliers. Data is an average of 2 biological replicates. ($n = 4803$ CSR-1 targets except klp-7). **c, d** A genomic view of *klp-7* showing Ribo-seq (**c**) and 22G-RNAs (**d**), normalized reads for the strain with codon-optimized *klp-7* (*klp7_co*) in pink compared to WT strain in gray. Data is average of two biological replicates. **e** $\log_2$ fold-change in expression of *csr-1*, *glh-4*, and *klp-7* upon *csr-1* RNAi compared to control RNAi by qPCR in the WT strain and *klp-7_co* (strain with *klp-7* codon optimization). Data are represented as mean ± SD for three biological replicates. Source data are provided as a Source Data file.

number of maternally inherited CSR-1 target mRNAs in somatic blastomere in the embryos[32]. Our observations also suggest a catalytic-independent function of CSR-1 in preventing piRNA-dependent chromatin silencing. Specifically, we showed that in the absence of CSR-1 protein, a different subset of CSR-1 target genes mostly comprised oogenic genes is misrouted into the piRNA pathway, which represses their expression at the transcriptional levels through the nuclear Argonaute HRDE-1. Therefore, in addition to the post-transcriptional regulation of germline mRNAs[25], CSR-1 can also license the transcription of germline genes, which was hypothesized previously based on transgene analysis[18,19] and shown directly here (Fig. 7). Mutation in the CSR-1 pathway was also shown to cause changes in the epigenetic landscape[52,53]. Given that HRDE-1 is known to promote the deposition of histone modifications associated with gene silencing, the effects observed upon mutation in components of the CSR-1 pathway might be the results of CSR-1 anti-silencing function. The majority of CSR-1 protected genes include oogenic

genes, which initiate their transcription during the developmental stage analyzed in this study. We predict that more oogenic genes might be protected by CSR-1 at a later time point but is difficult to study due to developmental defects accumulated at these later timepoints[25]. The CSR-1 protected targets do not overlap significantly with CSR-1 sliced targets. We, therefore, propose that CSR-1 slicer and anti-silencer function co-exist to regulate different germline gene expression programs. This regulation is also potentially spatially compartmentalized due to different functions of CSR-1 in P granules and cytosol, as discussed in another paragraph below. Further studies are required to uncouple the impact of the slicer and anti-silencer function using tools depleting CSR-1 spatially and temporally. CSR-1 is not the only mechanism that might license germline mRNAs. Indeed, apart from CSR-1 targeting of mRNAs[16], other mechanisms have been proposed to protect germline mRNA from piRNA silencing, including PATCs sequences in introns[17], and not yet completely defined features in coding sequence[15]. Therefore, that might also

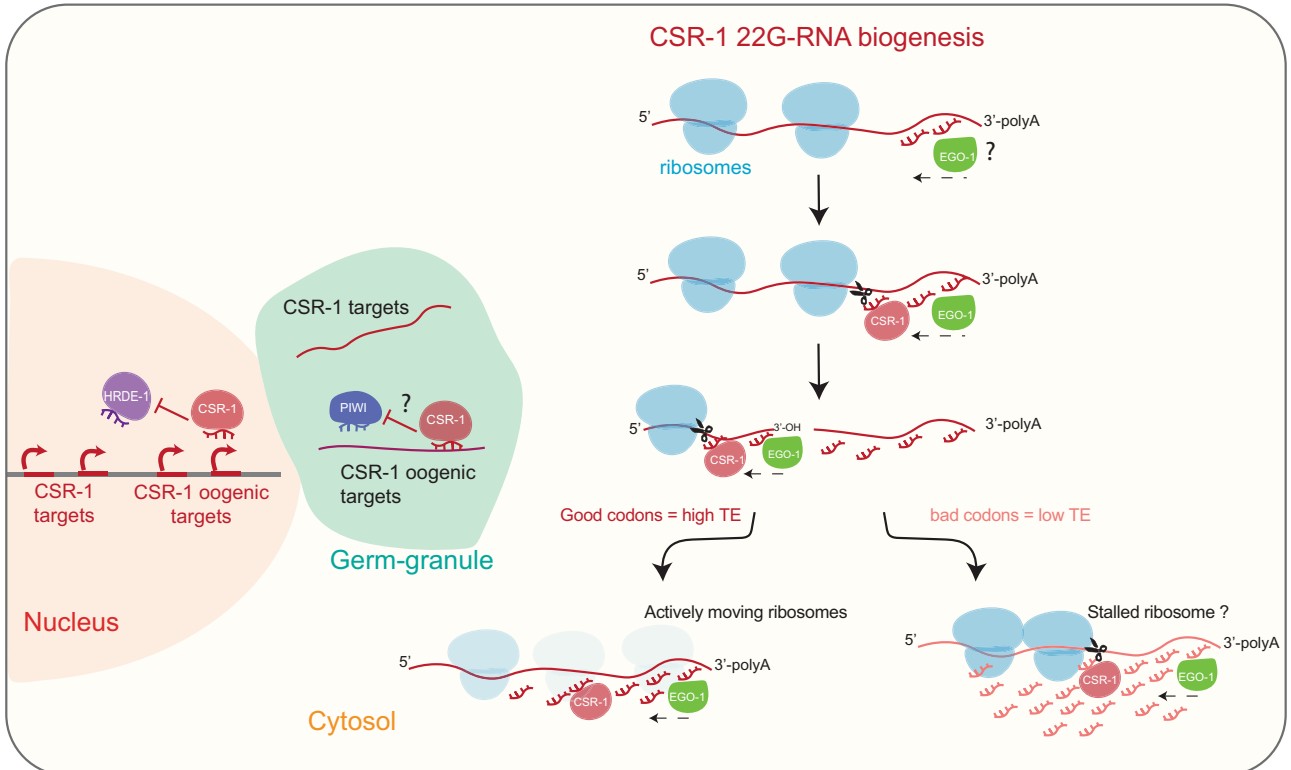

**Fig. 7 Model illustrating biogenesis of CSR-1-22G-RNA in the cytosol.** CSR-1 targets most of the germline-expressed genes. CSR-1 22G-RNAs are produced from mRNAs engaged in translation in the cytosol. We propose EGO-1 initiate 22G-RNA biogenesis at the 3′UTR on every actively translating mRNAs or by being recruited on specific 3′UTR sequences by yet unknown mechanism. However, to produce 22G-RNAs on coding sequence, CSR-1 slicing activity is required on the mRNA template. Codon usage and translation efficiency antagonistically regulate levels of 22G-RNAs production on different CSR-1 targets. We propose that CSR-1 can interact with ribosomes and the slicing event is more biased downstream of a possible stalled ribosome occupying a non-optimal codon site. CSR-1 slicer activity can regulate gene expression of few top targets, which further depends on the 22G-RNA levels. Additionally, CSR-1 can protect a set of mainly oogenic genes from piRNA-mediated HRDE-1 transcriptional silencing in a catalytic-independent manner.

explain why the removal of CSR-1 does not affect a large number of mRNAs.

**Biogenesis of CSR-1 22G-RNAs.** In this study, we have now established that the majority of Argonaute CSR-1 slicing activity cleaves target mRNAs to trigger the generation of RdRP-dependent 22G-RNAs on the gene body. We propose that CSR-1 slicer activity is required to generate new 3′-OH ends along the gene transcript to facilitate the initiation of 22G-RNA synthesis by RdRP (EGO-1) towards the 5′-end of the mRNA target. This is consistent with previous in vitro RdRP analysis showing that non-polyadenylated 3′-OH ends of RNAs served as better substrates for 22G-RNA synthesis[24], suggesting that the cleavage of RNA may be vital for the processivity of RdRPs. Based on these results, we speculate that no primary small RNAs are required to generate CSR-1 22G-RNAs along the mRNA sequence. Instead, CSR-1 catalytic activity triggers the synthesis of 22G-RNAs by the RdRP EGO-1, starting from the 3′UTR of the target transcripts. Even if the catalytic activity of CSR-1 is required to generate 22G-RNAs along the gene body of target transcripts, it is still unknown what triggers the recruitment of EGO-1 on the 3′UTR. Primary small RNAs, which are yet to be identified, might prime the activity of EGO-1. Alternatively, EGO-1 might produce low levels of 22 G-RNAs from the polyadenylated tail of mRNAs instead of cleaved 3′OH end products. Thus, these low levels of 22G-RNAs, which are then loaded into CSR-1, can initiate the production of 22G-RNAs along the gene body. RNA binding proteins and/or other unknown factors together with specific sequences in the

3′UTR might also recruit and initiate EGO-1-dependent 22G-RNAs from the 3′UTR of selected mRNAs.

**The role of translation and codon usage in CSR-1 22G-RNA biogenesis.** Germ granules are thought to be the site for all 22G-RNA synthesis and have been shown to be essential for the synthesis of piRNA-dependent 22G-RNAs. Whether CSR-1 22G-RNAs are also generated in germ granules is still unknown. In our current study, we show that CSR-1 22G-RNAs are synthesized in phase with ribosomes on actively translating mRNAs. P granules are known to be depleted of translating mRNAs, and P granule enriched mRNAs become translationally active upon P granule exit[42]. In addition, biochemical and proteomic characterizations of other cytoplasmic granules such as P bodies also show that those granules are depleted of ribosomal proteins[54]. We further showed that both CSR-1 and RdRP, EGO-1, are present in the polysome fractions, indicative of their interaction with translating mRNAs. We also observed a characteristic three-nucleotide periodicity between the start position of CSR-1-associated 22G-RNAs and RPFs, typical of the ribosomal footprint. A similar three-nucleotide periodicity has been observed for other co-translational events like 5′ to 3′ exonucleolytic decay of decapped mRNAs[55].

In contrast to CSR-1, we found PIWI was not enriched in polysome fractions, and downstream argonaute HRDE-1-bound piRNA-dependent 22G-RNAs are randomly distributed with respect to 5′end of the RPF, indicating that the results obtained with CSR-1 22G-RNAs are not due to sequence bias. We also show that RNAi of P granule components, which disrupt germ

granules and CSR-1 granule localization, results in the impairment of piRNA-dependent 22G-RNAs but not CSR-1-associated 22G-RNAs. Taken together, these results allowed us to conclude that the majority of CSR-1 22G-RNA biogenesis occurs in the cytosol co-translationally.

The co-translational synthesis of CSR-1 22G-RNAs raises the question of how CSR-1-22G-RNA biogenesis machinery is able to cope with the presence of ribosomes on the target transcripts. We show that non-optimal codons in germline mRNAs enhance the capacity of CSR-1 to prime the synthesis of EGO-1-dependent 22G-RNAs along the gene body. In fact, the translation efficiency of CSR-1 targets inversely correlates with 22G-RNA levels. We propose that the use of non-optimal codons by CSR-1 targets and priming of 22G-RNAs at stalled positions is a way to cope with the ribosomal presence on the target transcripts. Therefore, sequences that promote ribosome stalling promote targeting by CSR-1 to recruit EGO-1 on coding sequence to synthesize 22G-RNAs. To test this hypothesis, we have shown that the substitution of non-optimal codons with optimal codons is sufficient to allow germline mRNAs to escape CSR-1-dependent regulation. However, it is still unclear how EGO-1 initiates the synthesis of 22G-RNAs at the 3′-end of RPF, and this requires further investigation. One possibility is that CSR-1 and EGO-1 might coordinate their activity with the Ski complex, which extracts mRNA from 80 S ribosomal complexes in a 3′→5′ direction facilitating exosomal degradation[56]. Additionally, ribosome-phased endonucleolytic cuts possibly produced by the ribosome by the process called ribothrypsis, at the exit site of the mRNA ribosome channel may facilitate EGO-1 movement on transcript occupied by stalled ribosome[57]. There is increasing evidence that the translation machinery associates with the Argonautes and small-RNA biogenesis factors. Ribosome movement on translating mRNAs resolves mRNA structure to provide accessibility to Argonaute AGO2 downstream of the ribosome and promote AGO2-target interaction[58,59]. Another report showed that RNAi can occur co-translationally with an accumulation of ribosomes upstream of the dsRNA targeted region[60]. Ribosomes have been shown to coordinate with piRNA biogenesis factors in mouse testes to achieve endonucleolytic cleavage of non-repetitive long RNAs to produce pachytene piRNAs[61]. In plants, 22-nt siRNAs can repress translation, leading to induction of transitive small-RNA amplification by RNA-dependent RNA polymerase 6 (RDR6)[62]. Another recent report in plants showed that microRNA targeting recruits a double-strand RNA binding protein, which induces ribosome stalling, and the ribosome stalling enhances the generation of secondary small RNAs[63]. Therefore, we propose that the regulation of small-RNA biogenesis by ribosome occupancy and codon usage of the target transcript might be a general strategy adopted across evolution.

## Granule vs. cytosolic functions of CSR-1

We found that the slicer activity of CSR-1 and 22G-RNA biogenesis at germline mRNA targets are independent of P granules. This raises the question on the function of CSR-1 in germline granules. CSR-1 might be enriched in P granules of adult gonads to prevent CSR-1 slicer activity on the majority of germline mRNAs. Indeed, only 7.7% of CSR-1-dependent 22G-RNA targets are significantly regulated by CSR-1 slicer activity in adults. Moreover, the majority of these targets are CSR-1-interacting proteins, suggesting a negative feedback regulation of the CSR-1 pathway. This is in contrast with the recently described function of the maternally delivered CSR-1 in the embryo, which exclusively localizes in the cytosol of the somatic blastomere, where it cleaves and clears hundreds of maternal mRNA targets[32]. Therefore, we propose that CSR-1 slicer activity on mRNA targets is partially suppressed in the germline by titrating away a part of CSR-1 in P granules and primarily serves to generate interacting small RNAs in the cytosol that fully operates intra-generationally in the embryo. This also explains why despite targeting almost all germline genes, CSR-1 catalytic activity regulates the expression of only a few in the germline. In addition, CSR-1 localization in the P granule might serve to antagonize piRNA-dependent targeting on germline mRNAs and therefore license those transcripts to be translated in the cytosol. Indeed, we have shown that most of the piRNA-dependent 22G-RNAs are generated in P granules, and we propose that the competition between CSR-1 and PIWI might occur in P granules.

## Methods

### C. elegans strains and maintenance

Strains were grown at 20 °C on NGM plates seeded with E. coli OP50 using standard methods[64] unless otherwise stated. The wild-type reference strain used was Bristol N2. A complete list of strains used in this study is provided in Supplementary Data 5.

### Generation of CRISPR–Cas9 lines

Cas9-guide RNA (gRNA) ribonucleoprotein complexes were microinjected into the hermaphrodite syncytial gonad as described previously[65], and gRNA design and in vitro synthesis were done following the protocol detailed in[66]. For the introduction of a csr-1(D769A) mutation[66] in 3×flag::ha::csr-1 animals, we used a single-stranded oligonucleotide repair template ordered from IDT as standard 4 nM ultramer oligo. For the endogenous klp-7 gene replacement, we used two gRNAs, each one respectively targeting a region at the 5′ and 3′ of the klp-7 isoform b gene. A PCR repair template containing 33 bp homology arms was directly amplified from a plasmid containing a codon-optimized version of klp-7 (klp-7_co synthetic gene) synthesized from GenScript (Supplementary Data 6).

Mix concentrations were adapted from[67]. In brief, 10 μL mixes typically contained the following final concentrations: 0.1 μg/μL Cas9-NLS protein (TrueCut V2, Invitrogen), 100 ng/μL in vitro transcribed target-gene gRNA, 80 ng/μL of target-gene ssODN repair template or 300 ng/μL target-gene double-stranded DNA repair template and 80 ng/μL pRF4 (roller marker). Cas9 and the target-gene gRNA were pre-incubated 10–15 min at 37 °C before the addition of the other components to the mixture. dsDNA repair templates were subjected to a melting/annealing step[67] before addition to the final mix. A detailed list of gRNAs, single-stranded DNA, and double-stranded DNA repair templates and primers used for genotyping are provided in Supplementary Data 6.

### RNAi

RNAi clones for ego-1 and csr-1 used in this study were obtained from the Ahringer library[68]. For quadruple P granule RNAi (pgl-1, pgl-3, glh-1, and glh-4), pDU49 clone (gift from Updike lab[40]) was used. An empty vector (L4440) was used as a control in all of our RNAi experiments. RNAi experiments were performed by growing a synchronous population of L1 larvae on Petri dishes with NGM and IPTG (15 cm) seeded with concentrated RNAi food. For csr-1 and ego-1 RNAi, worms were grown from L1 to late L4 stage on RNAi food at 20 °C. For P granule RNAi, worms were grown for two generations at 25 °C[40]. Post-RNAi treatment, worms were harvested and sorted on COPAS biosorter to enrich late L4 larvae. RNAi efficacy was confirmed by RT-qPCR.

### ego-1 RNAi and auxin-induced CSR-1 degradation

For ego-1 RNAi worms were grown from L1 to 38 h post-hatching on RNAi or control food on IPTG containing plates and then washed twice with M9 buffer and then shifted to either Auxin plates or Ethanol plates (containing 500 μM auxin, 0.5% Ethanol or only 0.5% Ethanol respectively) to deplete degron-tagged CSR-1 by auxin-induced degradation as described before[32]. Plates were seeded with respective ego-1 RNAi or control RNAi food. Auxin-induced degradation was performed for 6 h. Worms were then harvested, washed with M9 buffer, and sorted on COPAS biosorter to enrich for Late L4 larval population. CSR-1 depletion was confirmed by live imaging.

### CSR-1 expression recovery post-auxin-induced degradation

A synchronous population of degron-tagged CSR-1 strain was grown on NGM plates containing 500 μM auxin, 0.5% ethanol from L1 to 38 h post-hatching to degrade degron-tagged CSR-1. After 38 h, worms were washed thrice with M9 buffer and divided into three parts. 1/3rd worms were immediately sorted on COPAS biosorter to enrich for a synchronous population for 0 h recovery time point of CSR-1 expression. The rest of the worms were seeded on two NGM plates and allowed to grow in the absence of auxin induction for 5 or 10 h to recover CSR-1 expression. Worms were washed with M9 buffer at respective time points and sorted using COPAS biosorter to enrich for a synchronized population for each time point. CSR-1 expression was monitored using live imaging.

**Brood-size assay**. For the brood size, single L1 larvae were manually picked and placed onto NGM plates seeded with *E. coli* OP50 and grown at 20 °C or 25 °C until adulthood and then transferred on a new plate every 24 h for a total of 2 transfers. The brood size of each worm was calculated by counting the number of embryos and larvae laid on the three plates. Embryonic lethality was measured by counting the number of the unhatched embryo (dead) 24 h post laying compared to total embryos laid.

**Counting of oocytes in population**. For the WT (N2) and CSR-1 catalytic mutant, germlines of adult worms (72 h post-hatching) were dissected and stained with DAPI, and the number of oocytes was counted.

**Sorting**. Large populations of the Late L4 larvae stage from the synchronized population were sorted using the COPAS BIOSORT instrument (Union Biometrica), according to the manufacturer's guidelines. The population was sorted using two size parameters, Time of flight (TOF) and extinction. The stage of the sorted population was validated by counting worms under a microscope by scoring features like closed vulva and absence of oocytes as a characteristic of late L4 stage larvae. First-generation homozygotes for CSR-1 KO or CSR-1 ADH were sorted by excluding GFP-positive heterozygote worms. *klp-7_co* heterozygote strain was sorted using GFP marker, and GFP-positive worms were sorted.

**Imaging**. Transgenic worms were mounted on 2% agarose pads in a drop of M9 with 1 mM Levamisole. Images were acquired on ZEISS LSM 700 microscope with a ×40 objective or ×63 objective for the PGL-1::mCardinal; ZNFX-1:: TagRFP; MUT-16::GFP, DEPS-1::GFP; mCherry::CSR-1; GLH-1::GFP;mCherry::CSR-1 and GLH-1::GFP; mCherry::CSR-1 ADH. Images were acquired using the ZEISS ZEN software and processed using ImageJ v.2.0.0. mCherry::CSR-1 in *mut-16* and *znfx-1* mutant background were imaged on Zeiss Axio Imager M2 and were acquired using MetaMorph software. All strains are listed in Supplementary Data 5. For counting oocytes, dissected gonads were mounted in DAPI containing Vectashield mounting medium, and oocytes were counted by visualizing on Zeiss Axio Imager M2.

**Western blotting**. Worms were lysed in 1x NuPAGE LDS sample buffer (ThermoFisher Scientific) and heated at 90 °C for 10 min. Any debris was removed by centrifuging at $18,000 \times g$. ~50 μg of protein extracts was then resolved on precast NuPAGE Novex 4–12% Bis-Tris gels (Invitrogen, NP0321BOX). The proteins were transferred to a nylon membrane with the semidry transfer Pierce Power System (ThermoFisher Scientific) using the pre-programmed method for high-molecular-mass protein. The primary antibodies used included anti-KLP-7[25] (1:1000 dilution) (a gift from the Desai laboratory), anti-tubulin (Ab6160, Abcam) (1:1000 dilution), anti-GAPDH (Ab125247, Abcam) (1:2000 dilution), anti-PGL-1[69] (1:2000 dilution) (a gift from the Strome laboratory), anti-PRG-1[70] (1:2000 dilution) (a gift from the Mello laboratory), anti-Flag (F3165, Sigma) (1:1000 dilution), anti-RPS-3 (ab128995, Abcam) (1:3000 dilution) and the secondary antibodies used included anti-rabbit (31460, Pierce) (1:10000 dilution), anti-mouse (31430, Pierce) (1:10000 dilution) and anti-rat (A9037, Sigma) (1:10000 dilution) HPR antibodies. The SuperSignal West Pico PLUS Chemiluminescent Substrate was used to detect the signal using a ChemiDoc MP imaging system (Biorad).

**RNA extraction**. For total RNA extraction, synchronous and sorted populations of ~1000 worms as described for individual experiments were frozen in dry ice with TRIzol™ (Invitrogen, Ref. 15596026). After five repetitions of freeze and thaw, total RNA was isolated according to the manufacturer's instructions. For RNA extraction after IP, TRI Reagent was directly added to beads, and RNA extraction was performed as per the manufacturer's instructions. For RNA used for RNA-seq or RT-qPCR, DNase treatment was performed using a maximum of 10 μg RNA treated with 2U Turbo DNase (Ambion) at 37 °C for 30 min followed by acid phenol extraction and ethanol precipitation. An Agilent 2200 TapeStation System was used to evaluate the RIN indexes of all of the RNA preps, and only samples with RNA integrity numbers (RIN) > 8 were used for downstream applications.

**Quantitative reverse transcription PCR (RT-qPCR)**. Reverse transcription was performed according to manufacturer's instructions using M-MLV reverse transcriptase (Invitrogen, Ref. 28025013), and qPCR was performed using Applied Biosystems Power up SYBR Green PCR Master mix following the manufacturer's instructions and using an Applied Biosystems QuantStudio 3 Real-Time PCR System and analyzed using QuantStudio™ Design and Analysis software V 2.2. Primers used for qPCR are listed in Supplementary Data 7.

**IP/total- sRNA-seq**. Total RNA from at least 1000 sorted worms with RIN > 9 was used to generate small-RNA libraries. For 22G-RNAs from IP, IP was performed using ~10,000 synchronized and sorted worms for FLAG-CSR-1 or ~70,000 for GFP-HRDE-1. Worms were lysed in small-RNA IP buffer (50 mM HEPES pH 7.5, 500 mM NaCl, 5 mM MgCl$_2$, 1% NP-40, 10% glycerol, 1x Halt protease inhibitors and RNaseIN 40 U/mL), using a chilled metal dounce. Crude lysates were cleared of debris by centrifuging at $18,000 \times g$ at 4 °C for 10 min. Ten percent of the extract

was saved as input, and total RNA was extracted using TRIzol™ as described above. The rest of the extract was incubated with 30 μl of Anti-FLAG M2 Magnetic Agarose Beads suspension (Sigma M8823) or 25 μl GFP-Trap Magnetic Agarose (Chromotek gtma-10) for FLAG-CSR-1 or GFP-HRDE-1 respectively, for 1 h at 4 °C. After four washes of the beads with the small-RNA IP buffer, the RNA bound to the bait was extracted by adding TRIzol™ to beads as described above. The library preparation was performed essentially as described previously[66]. Amplified libraries were multiplexed to purify further using PippinPrep DNA size selection with 3% gel cassettes and the following parameters for the selection: BP start (115); BP end (165). The purified libraries were quantified using the Qubit Fluorometer High Sensitivity dsDNA assay kit (Thermo Fisher Scientific, Q32851) and sequenced on a NextSeq-500 Illumina platform using the NextSeq 500/550 High Output v2 kit 75 cycles (FC-404-2005).

**IP and radiolabeling of sRNA**. IP was performed as described above for FLAG::CSR-1 and FLAG::CSR-1 ADH. Ten percent of total extract and IP was processed for western blotting. The rest of the IP was used for extracting RNA using TRIzol as described above. RNA was treated with polyphosphatase to generate monophosphate 5′ends[66]. 5′end of RNA from the above step was labeled using γP$^{32}$-ATP using T4-polynucleotide kinase (EK0031, Thermofisher Scientific) as per manufacturer's instructions. Labeled RNA was purified using 1.8x SPRI beads with isopropanol and resuspended in 10 μL water. TBE Urea loading buffer (Thermofisher Scientific) was added to the sample, and RNA denatured at 70 °C for 5 min and then resolved on Novex™ 15% TBE Urea gel (Thermofisher Scientific). The resolved gel was exposed on a Phosphor screen and scanned on Typhoon FLA 9000 scanner.

**Gro-seq**. One thousand synchronized and sorted Late L4 worms for WT (N2), *csr-1* catalytic mutant and *csr-1* KO were collected as described above. Nuclear Run-on reaction was performed by incorporating 1 mM Bio-11-UTP, followed by RNA extraction and biotinylated nascent RNA enrichment as described previously[32]. Libraries were prepared by repairing 5′-OH of fragmented RNAs by Polynucleotide Kinase (Thermo scientific), followed by 3′ and 5′ adapter ligation as described previously[32]. Adapter ligated RNA was reverse transcribed using SuperScript IV Reverse Transcriptase (Thermo Fisher Scientific) following manufacturer conditions, except that reaction was incubated for 1 h at 50 °C. cDNA was PCR amplified with specific primers using Phusion High fidelity PCR master mix 2x (New England Biolab) for 18–20 cycles. Libraries were analyzed on Agilent 2200 TapeStation System using high-sensitivity D1000 screentapes and quantified using the Qubit Fluorometer High Sensitivity dsDNA assay kit (Thermo Fisher Scientific, Q32851). Multiplexed libraries were sequenced on a NextSeq-500 Illumina platform using the NextSeq 500/550 High Output v2 kit 75 cycles (FC-404–2005).

**Strand-specific RNA-seq library preparation**. DNase-treated total RNA with RIN > 8 was used to prepare strand-specific RNA libraries. Ribosomal and mitochondrial rRNAs were depleted using a custom RNAse-H-based method to degrade rRNAs using complementary oligos as described previously[66].

Strand-specific RNA libraries were prepared using at least 100 ng of rRNA depleted RNAs using NEBNext Ultra II Directional RNA Library Prep Kit for Illumina (E7760S). RNA libraries were analyzed on Agilent 2200 TapeStation System using high-sensitivity D1000 screentapes and quantified using the Qubit Fluorometer High Sensitivity dsDNA assay kit (Thermo Fisher Scientific, Q32851). Multiplexed libraries were sequenced on a NextSeq-500 Illumina platform using the NextSeq 500/550 High Output v2 kit 75 cycles (FC-404–2005).

**Ribo-seq**. Ribo-seq has been performed as described previously[43] with some modifications[32]. Briefly, 10,000 late L4 worms were sorted using COPAS biosorter as described above and were lysed by freeze grinding in liquid nitrogen in Polysome buffer (20 mM Tris-HCl pH 8, 140 mM KCl, 5 mM MgCl$_2$, 1% Triton X-100, 0.1 mg/mL cycloheximide) and ~1 mg extract was digested by RNase I (100 U) at 37 °C for 5 min. Debris was clarified by centrifuging at $18,000 \times g$ followed by fractionation on a discontinuous sucrose gradient (10–50%) by ultracentrifugation at $260,110 \times g$ for 3 h in an SW41-Ti rotor (Beckman coulter). Monosome fractions were collected by pumping Fluorinert FC-40 and using a fraction collector by measuring UV trace. RNA extracted from the monosome fraction was DNase treated as described above and fragments of 28–30 nucleotides were size selected by resolving on a 15% TBE-Urea gel. 3′phosphate was removed (PNK buffer pH 6.5 (70 mM Tris pH 6.5, 10 mM MgCl$_2$, 1 mM DTT), T4 PNK (Thermo Scientific), RNaseIN 40 U/mL, 20% PEG400) and 5′-end was phosphorylated by treating RNA with T4-Polynucleotide Kinase (1x PNK buffer (Thermo Scientific), 1 mM ATP). In all, 28–30 nucleotide Ribosome-protected fragments (RPF) were then cloned with the sRNA-seq library preparation approach, as described previously[32,66].

**Polysome profiling and blot**. Lysates were fractionated on a discontinuous sucrose gradient (10–50%) as described above, with the exception that no RNase treatment was performed. Twenty-two fractions were collected by pumping of Fluorinert FC-40 and using a fraction collector while simultaneously measuring the UV trace. Fractions were precipitated with 10% TCA at 4 °C for 4 h and centrifuged at

18,000 × g for 10 min. Pellets were washed with pre-chilled acetone twice, followed by resuspension in 2x NuPAGE LDS sample buffer (ThermoFisher Scientific) and heated at 90 °C for 10 min. Samples were processed for western blotting as described above.

**Immunoprecipitation-mass spectrometry (IP-MS/MS).** IPs for the MS/MS analysis were performed as described previously[66]. Briefly, a synchronous population of 120,000 (for CSR-1 IPs for RNase treatment or control condition and PRG-1 IPs in RNase and no RNase condition) worms were harvested at 48 h post-hatching or 20,000 (for CSR-1 IPs comparing WT IP with catalytic mutant) worms were harvested and sorted at 44 h post-hatching and lysed by using a chilled metal dounce in the IP buffer (50 mM HEPES pH 7.5, 300 mM NaCl, 5 mM MgCl$_2$, 10% Glycerol, 0.25% NP-40, protease inhibitor cocktails (Fermentas). Crude lysates were cleared of debris by centrifuging at 18,000 × g at 4 °C for 10 min. For RNase treatment, RNase I (Invitrogen) 50 U/mg of the extract was used at 37 °C for 5 min. Approximately 5 mg of protein extract (for CSR-1 IPs and PRG-1 IPs in RNase or control condition) or 1 mg of protein extract (for CSR-1 IPs comparing WT IP with catalytic mutant) was incubated with 15 μl of packed Anti-FLAG M2 Magnetic Agarose Beads (Sigma M8823) for 1 h at 4 °C. After four washes with the IP buffer, the beads were washed twice with 100 μL of 25 mM NH$_4$HCO$_3$. Finally, beads were resuspended in 100 μL of 25 mM NH$_4$HCO$_3$ and digested by adding 0.2 μg of trypsin/LysC (Promega) for 1 h at 37 °C. Samples were then loaded into a homemade C18 Stage Tips for desalting (principally, by stacking one 3 M Empore SPE Extraction Disk Octadecyl (C18) and beads from SepPak C18 Cartridge Waters into a 200 μl micropipette tip). Peptides were eluted using a ratio of 40:60 MeCN: H$_2$O + 0.1% formic acid and vacuum concentrated to dryness. Peptides were reconstituted in injection buffer (2:98 MeCN: H$_2$O + 0.3% TFA) before nano-LC-MS/MS analysis as described previously[66].

## Data analysis

*Sequencing data analyses.* Multiplexed data were demultiplexed using Illumina bcl2fastq converter version v2.17.1.14. Analysis for RNA-seq, sRNA-seq and GRO-seq have been performed as previously described[32,66]. Quality control was performed with fastQC version v0.11.5. HISAT2 version 2.0.4 was used for mapping RNA-seq data. Bowtie2 version 2.3.4.1 was used for all other sequencing data. Unless otherwise stated, computations were done using Python and UNIX utilities, either as standalone scripts or as steps implemented in a Snakemake[71] workflow. The scripts and workflows are available at https://gitlab.pasteur.fr/bli/bioinfo_utils. For mapping 22G-RNA with 3′ polyuridinylation, among the small RNA reads that initially did not map, those starting with G followed by 20 to 25 nucleotides and then one or more Ts were selected, and their T-tail was trimmed. Those reads were then re-mapped in the same way as initially and classified using the same criteria as other small RNAs[66]. If classified as "22 G" by this procedure, they were actually considered "poly-U 22 G" or "siu 22 G".

For Ribo-seq data (data analysis pipeline available at the same address), the analysis was performed according to the following steps. The 3′ adapter was trimmed from raw reads using Cutadapt v.1.18[72] using the following parameter: -a TGGAATTCTCGGG TGCCAAGG –discard-untrimmed. The 5′ and 3′ UMIs were removed from the trimmed reads using cutadapt with options -u 4 and -u -4. After removing UMIs, the reads from 28 to 30 nt were selected using bioawk (https://github.com/lh3/bioawk, git commit fd40150b7c557da45e781a999d372abbc634cc21).

The selected 28–30-nucleotide reads were aligned to the *C. elegans* genome sequence (ce11, *C. elegans* Sequencing Consortium WBcel235, with an added extra chromosome representing the codon-optimized klp-7 for some libraries) using Bowtie2[73] v.2.3.4.3 with the following parameters: -L 6 -i S,1,0.8 -N 0.

Reads mapping on sense orientation on annotated protein-coding genes were considered as Ribosome-protected fragments (RPF). Such reads were extracted from mapping results using samtools[74] 1.9 and bedtools[75] v2.27.1. RPF reads of size 29 were further classified into subcategories, based on the codons found at the positions corresponding to the A (16–18 nt) and P (13–15 nt) sites of the ribosome. Codon optimality was defined as explained below. Those reads were re-mapped on the genome using bowtie2 (version 2.3.4.3) with options -L 6 -i S,1,0.8 -N 0. The resulting alignments were used to generate bigwig files using a custom bash script using bedtools version v2.27.1, bedops[76] version 2.4.35, and bedGraphToBigWig version 4. Read counts in the bigwig file were normalized by million "non-structural" mappers, that is, reads of size 28 to 30 nt mapping on annotation not belonging to the "structural" (tRNA, snRNA, snoRNA, rRNA, ncRNA) categories, and counted using featureCounts[77] v.1.6.3. These bigwig files were used to generate "metaprofiles" where normalized coverage information (RPM for reads per million) was averaged across replicates and represented along sets of selected genes. The metaprofiles were generated using a Python script based on the deepTools[78] and gffutils (https://github.com/daler/gffutils) libraries. Translation efficiency was calculated as the ratio of TPMs of Ribo-seq and RNA-seq.

*Distance distribution analyses.* The distribution of the distances between re-mapped RPF and 22G-RNA-seq reads was computed by counting distances between 5′-ends of RPF and 22G-RNA reads of opposite strandedness, only considering 22G-RNA reads within a distance of +/− 120 bp from the RPF read and only considering RPF reads mapping in the sense direction within the coordinates of a gene among a selected list. Counts were transformed into z-scores using the Scipy[79] library (version 1.3.2). A plot

of distance distribution, within the (−15, 45) distance range, was made using the Matplotlib library (https://doi.org/10.1109/MCSE.2007.55) version 3.1.1. This was done using z-scores in order to have comparable values between different combinations of libraries. A plot of dominant periods in distance distribution signal was made using the Matplotlib library (https://doi.org/10.1109/MCSE.2007.55) version 3.1.1. The dominant periods were obtained using the fast Fourier transform function of the Scipy library (version 1.3.2)[79]. This was done using z-scores in order to have comparable values between different combinations of libraries.

*Read composition analyses.* Reads of a class of interest (small RNAs, or size-restricted subclasses thereof, RPF or RPF subclasses) and found mapping on a given gene, either on the CDS or on the first 100 bp of the 3′UTR and either mapping in sense or antisense with respect to the gene annotation, were gathered from their re-mapping results using custom Python code based on pysam (https://github.com/pysam-developers/pysam) a Python wrapper for the samtools package[74]. For a given such set of reads, the proportions of each nucleotide at each position starting from the 5′end or from the 3′-end of the reads were computed. These proportions were averaged across a set of genes of interest and represented as a stacked barplot, where each stack corresponds to a position in the reads, with the most frequent nucleotides on top. For comparing read composition of sRNA from CDS and 3′UTR, to account for overall genome composition variability across genes or gene features, "scaled" proportions were computed by dividing the nucleotide proportions by those found in the genomic region on which the reads were found mapping (CDS or first 100 bp of 3′UTR of a given gene), then similarly averaged across a set of genes of interest and represented as a stacked barplot. The core functionalities used in these analyses are implemented in the following Python library: https://gitlab.pasteur.fr/bli/libreads. The whole code we used is available upon request.

*Analysis of codon usage.* All protein-coding genes were categorized based on their Translation Efficiency in the following categories Log$_2$TE ≥ 3, ≥2, ≥1, ≤−1, ≤−2, and ≤−3. Relative synonymous codon usage was calculated for genes in each category using the CAI calculator[80]. To calculate enrichment of codons usage in each of the categories, differential RSCU of respective categories of genes was calculated by normalizing their RSCU with RSCU of genes showing a TE of ~1 (Log$_2$TE 0 ± 0.1). Codons enriched in highly translated mRNAs (Log$_2$TE ≥ 3) were considered optimal codons, and codons that were avoided were considered non-optimal. Similarly, differential RSCU analysis was performed for CSR-1 targets.

*Gene ontology and enrichment analysis.* Gene ontology was performed using WormCat tool[81]. Enrichment was calculated using the webtool http://nemates.org/MA/progs/overlap_stats.html.

*tRNA copy number and TPM.* tRNA copy number was determined using tRNAscan-SE 2.0[82]. TPMs for the tRNAs were extracted from the GRO-seq dataset from WT late L4-staged worms.

*Determination of a codon-optimized sequence for klp-7.* The codon-optimized sequence for *klp-7* was computed with a Python script using BioPython[83] as follows: To each amino acid, a corresponding optimal codon was associated based on a given optimality ranking. Here, the codon ranking was based on usage in highly translation efficient proteins, as explained above. Then, each codon in the CDS of the native *klp-7* gene was replaced with the optimal codon associated with the corresponding amino acid. For mapping purposes, the resulting sequence was added to the genome as if it was an extra chromosome, and the transgene was added to the annotation files used for read counting. In order to produce comparable bigwig tracks between libraries obtained on different strains (codon-optimized or not), a Python script based on the pyBigWig library (https://doi.org/10.5281/zenodo.594045) was used to relocate the values on the extra chromosome to the actual genomic position of *klp-7*.

*MS/MS data analysis.* For identification, the data were searched against the *C. elegans* (CAEEL) UP000001940 database (Taxonomy 6239 containing one protein sequence par gene) using Sequest HT through Proteome Discoverer (v.2.2). Enzyme specificity was set to trypsin, and a maximum of two missed cleavage sites was allowed. Oxidized methionine and N-terminal acetylation were set as variable modifications. Maximum allowed mass deviation was set to 10 ppm for mono-isotopic precursor ions and 0.6 Da for MS/MS peaks. The resulting files were further processed using myProMS v.3.9[84] (work in progress). False-discovery rate (FDR) was calculated using Percolator and was set to 1% at the peptide level for the whole study. Label-free quantification was performed using peptide extracted ion chromatograms (XICs), computed with MassChroQ v.2.2.1[85]. For protein quantification, XICs from proteotypic peptides shared between compared conditions (TopN matching) with missed cleavages were used. Median and scale normalization was applied on the total signal to correct the XICs for each biological replicate (N = 4). To estimate the significance of the change in protein abundance, a statistical test based on a linear model adjusted on peptides and biological replicates was performed, and P-values were adjusted using the Benjamini–Hochberg FDR. Proteins with at least three total peptides in all replicates, a twofold enrichment, and an adjusted P-value < 0.05 were considered

significantly enriched in sample comparisons. The MS proteomics data have been deposited to the ProteomeXchange Consortium via the PRIDE[86] partner repository with the dataset identifier PXD012557 and PXD020293.

*Statistics and reproducibility.* Almost all the experiments shown in this study were performed independently at least twice, and no inconsistent results were observed. IP and MS experiments were conducted with four biological replicates. Ribo-seq was performed using three biological replicates. All the RNA-seq experiments, GRO-seq, sRNA-seq, IP-sRNA-seq, were performed using two biological replicates. RT-qPCRs to test RNAi efficiency in samples for sequencing experiments were performed in their respective biological experiments. RT-qPCRs for gene expression changes otherwise were performed with at least three biological replicates. Most of the graphs were generated using GraphPad Prism 9. Log fold-changes for almost all the plots were calculated on the mean of biologically independent replicates. For details of the particular statistical analyses used, precise *P*-values, statistical significance, and sample sizes for all of the graphs, see the figure legends.

**Gene lists**. The gene lists generated in this study are provided in Supplementary Data 1 together with previously identified gene lists[14,22,23,25,66,87,88].

**Reporting summary**. Further information on research design is available in the Nature Research Reporting Summary linked to this article.

## Data availability

All sequencing data (GRO-seq, RNA-seq, and sRNA-seq from total lysate or IP experiments, Ribo-seq) are available at the Gene Expression Omnibus (GEO) under accession code GSE155077. The MS proteomics data have been deposited to the ProteomeXchange Consortium via the PRIDE partner repository with the dataset identifier PXD012557 and PXD020293. The data supporting the findings of this study are available from the corresponding authors upon reasonable request. Source data are provided with this paper.

## Code availability

Custom scripts of this study are available from the corresponding author on request. Custom code and data analysis workflows are available at https://gitlab.pasteur.fr/bli/bioinfo_utils.

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

## Acknowledgements

We would like to thank all the members of the Cecere laboratory, Manish Grover, Sudarshan Gadadhar, and Angela Anderson (Life Science Editors), for the helpful discussions on the paper. We thank Micheline Fromont for her help to set up Ribosome profiling. We thank Celine Didier for technical assistance. Sequencing was performed at the Biomics centre at the Institut Pasteur. We thank the Heng-Chi Lee lab, Miska lab, Desai lab, Strome lab, Mello lab, Updike lab, and Kennedy lab for sharing strains and reagents. Some strains were provided by the CGC, funded by the NIH Office of Research Infrastructure Programs (P40 OD010440). This project has received funding from the Institut Pasteur, the CNRS, and the European Research Council (ERC) under the European Union's Horizon 2020 research and innovation program under grant agreement No. ERC-StG- 679243. M.S. and E.C. were supported by the Pasteur-Roux-Cantarini Postdoctoral Fellowship program. P.Q. was supported by Ligue Nationale Contre le Cancer (SFB19032). F.D. and D.L. have received funding from Région Ile-de-France and Fondation pour la Recherche Médicale grants to support this study.

## Author contributions

G.C. and M.S. identified and developed the core questions addressed in the project. M.S. performed most of the experiments and analyzed the results together with G.C. E.C. and L.B. generated the strains used in this study with the help of M.S. and P.Q. E.C. performed the RNA-seq for CSR-1 mutants and IP-sRNA-seq of HRDE-1 in CSR-1 KO and phenotypic characterization of csr-1 mutants. P.Q. performed GRO-seq for CSR-1 mutants. B.L. performed all the bioinformatics analysis along with M.S. S.P. contributed for distance mapping analysis of 22G-RNA reads, and Ribo-seq reads with B.L. F.D. and D.L. performed MS/MS experiments and analyzed the data together with M.S. M.S. and G.C. wrote the paper with the contribution of all authors.

## Competing interests

The authors declare no competing interests.

## Additional information

**Peer review information** *Nature Communications* thanks Zissimos Mourelatos and other, anonymous, reviewers for their contribtuions to the peer review of this work. Peer review reports are available.

