## [Peer Review File · Nature Communications]

REVIEWER COMMENTS

Reviewer #1 (Remarks to the Author):

The Argonaute, CSR-1, is quite mysterious, in that it both protects transcripts from piRNA-mediated slicing and also itself slices target mRNAs. In this manuscript, Singh et al design a series of elegant experiments to address this long-standing enigma. First, they demonstrate that the slicing activity of CSR-1 is required to promote 22G-RNA biogenesis, but that only RNAs with very high 22G levels are down-regulated post-transcriptionally by CSR-1. They then further demonstrate that low translation efficiency promotes CSR-1 22G-RNA abundance and that increasing translation efficiency of a CSR-1 target disrupts 22G biogenesis and function. This work will certainly influence the way the field thinks about and understands the role of CSR-1.

Major comments –

Are there distinguishing features for the genes that are protected by CSR-1 from piRNA mediated silencing (Fig 1g-h)? For example, high or low levels of CSR 22G siRNAs? Why are only 227 genes falling into this class when protection from piRNA-mediated silencing seems to be a major theme of previous CSR-1 work?

Line 156 – The authors claim that ADH CSR-1 does not show any loss of 22G-RNA binding, however small RNA-seq would not necessarily give a quantitative answer to this question (both because PCR amplification is used for library generation and because analysis generally requires normalizing to library depth). By immunoprecipitating WT and ADH CSR-1, followed by radiolabeling of co-precipitating small RNAs, the authors could get a more accurate and quantitative assessment of how the ADH mutant affects small RNA binding.

In the auxin-mediated CSR-1 depletion (Fig 2d), why are CSR-1 siRNAs still present at the 3' ends of CSR-1 targets. Are these bound by low levels of CSR-1 (but not sufficient to promote cleavage/spreading) or are they bound by another Argonaute protein?

The knockdown of P granule components to demonstrate that CSR-1 22G-RNA biogenesis occurs in the cytoplasm is a beautiful experiment, but I see two caveats that should be discussed. First, some CSR-1 localization is still visible in the perinuclear space, so it remains possible that some 22G-RNA biogenesis is occurring here. Second, biogenesis of CSR class 22G-RNAs in the absence of P granules doesn't preclude the possibility that 22G-RNA biogenesis does occur in P granules when they are present, they are just not necessary.

I don't fully follow how Fig 4c-d show that the synthesis of CSR 22G-RNAs is from translating RNAs. Is the proposal that the RdRP is initiating 22G biogenesis just 5' or just 3' of a translating ribosome? Presumably RdRP initiation is limited to starting at a G, which would restrict the possible initiation sites. Is there a physical interaction between the RdRP and the ribosome? Similar question for 5f – how is a 22G siRNA synthesized within the ribosome protected region? Especially if the ribosome and RdRP are moving in opposite directions on the mRNA?

Minor comments –

How does this new list of CSR-1 target genes at L4 stage compare to previous CSR-1 target lists (Claycomb, 2009 for adults and Conine 2013 for adult males)?

Extended Data Fig. 2f is never referenced in the text. Also for this figure, I'm assuming that grey dots represent all CSR-1 target genes, but this is not explicitly stated in figure or figure legend.

Extended data Fig 6a does not seem like the right figure to demonstrate that 22G-RNA levels are

independent of mRNA abundance. Maybe a scatterplot of 22G abundance vs mRNA abundance would be a more clear way to make this point.

Reviewer #2 (Remarks to the Author):

The authors of this nicely written manuscript address the biogenesis of coding sequence derived antisense 22G-RNAs that are loaded in CSR-1 in the germline of *C. elegans*. This is an important area of investigation because unlike the CSR1 22G-RNAs that target repetitive elements, the biogenesis and function of 22G RNAs that target genic mRNA is a mystery. In this paper, the authors find that the biogenesis of CDS-derived antisense 22G RNAs is cotranslational and triggered by the cleavage activity of CSR1 itself in the vicinity of ribosomes that occupy suboptimal codons (and hence are slower). They show that such biogenesis occurs in the cytosol (as would be expected) and not in germ granules. By further investigating the differences between the molecular phenotypes of catalytically inactive CSR1 and CSR1 KO they identify a catalytic-independent function of CSR1, which is to prevent chromatin silencing by piRNAs. Overall, the study is nicely performed, the data and analyses are of high quality and support the authors' main conclusions.

Reviewer #3 (Remarks to the Author):

A study by Cecere and colleagues presents a comprehensive analysis of the effects of slicer activity on small RNAs that interact with the CSR-1 Argonaute protein, which suggest that it can act to promote biogenesis of small RNAs that induce mRNA decay when target mRNAs stall during translation. This paper addresses an apparent paradox regarding the multifunctional CSR-1 Argonaute protein, which has been previously shown to possess an 'anti-silencing' activity that promotes transcription of germline genes but also has the ability to slice and degrade some germline mRNAs, including those that regulate chromosome segregation during mitosis in embryos. The authors demonstrate that the RDRP EGO-1 promotes siRNA biogenesis in response to CSR-1 slicing of mRNAs, such that small RNAs homologous to the coding sequence mRNAs are depleted when EGO-1 is knocked down when CSR-1 is slicer-dead. The authors provide several lines of evidence that CSR-1-dependent coding sequence RNAs are phased such that they commonly begin at rare codons where ribosomes may transiently stall. They demonstrate that mRNA levels can be tuned by altering codon bias. The authors conclude that ribosomal mRNAs are likely to be targeted by CSR-1 slicer activity for cleavage and small RNA biogenesis in the cytoplasm, in part based on localization of CSR-1 by indirect immunofluorescence. I am uncertain about the strength of the conclusion that cytoplasmic CSR-1 is relevant to reducing mRNA, but many of the conclusions of this study are novel and interesting. The writing of this manuscript could also be improved in other places. I suggest revision of this manuscript by clarifying the background and significance of the data. It may also be helpful to re-frame the concept that the authors provide for CSR-1 primary siRNAs, in part by carefully examining small RNAs that are created at 3' UTRs.

Major comments:

1. It might be worth citing the original papers by Maine and Sundaram that identified CSR-1 and EGO-1 as part of a complex of proteins that functions together during development, the Maine paper pointing to a role in regulation of the epigenome that may not be understood yet. One possibility is that the effects observed by Maine and Sundaram are due to excess levels of mRNAs targeted by CSR-1 and EGO-1.
2. Because of the paradoxical nature of CSR-1 functions, it might be worth pointing out that although the original observations of Claycomb and Mello that the CSR-1/EGO-1 complex physically associates with condensed chromosomes, that this association is likely relevant to the anti-silencing role of CSR-

1 in promoting transcription. Then, you can clearly explain that slicer activity of CSR-1 has been previously linked to mRNA destruction of genes relevant to mitosis, which is distinct from a role for CSR-1 in transcriptional activation.

3. Line 67. 'Targeting by CSR-1 22G RNAs can function as an anti-silencing mechanism to protect germline mRNAs from piRNAs silencing . . . Germline mRNAs remain protected from piRNAs silencing even in the absence of CSR-1'. These sentences are confusing based on the paradoxical function of CSR-1 in regulating gene expression. The CSR-1 antisilencing mechanism does not protect germline mRNAs from piRNA silencing, it protects germline expressed genes from transcriptional silencing by piRNAs. As piRNAs promote transcriptional silencing, germline genes remain protected from piRNA silencing in the absence of the CSR-1 anti-silencing activity.

4. 'To what extent endogenous germline genes are regulated by the antagonistic functions of CSR-1 and piRNA pathways remains elusive'. Perhaps comment here about the rpm of CSR-1 small RNAs that promote anti-silencing of germline genes: is it known that they are rare?

5. 'Worms lacking CSR-1 protein display downregulation of germline CSR-1 targets, those expressing CSR-1 catalytic mutant show upregulation of its germline gene targets.' This is confusing. Does lack of CSR-1 only cause downregulation of germline CSR-1 targets? Does lack of CSR-1 not also result in upregulation of some targets (that are targeted by slicer activity)? It might be helpful to state here how many genes are known to be downregulated or upregulated by CSR-1, which will help the reader understand the significance of your advance.

6. 'However, disruption of a specific germ granule, the mutator foci'. The term 'Germ granules' is confusing in this manuscript. *C. elegans* germ granules are composed of at least three compartments that lie next to each other and be distinct but also may physically overlap. This needs to be clearly conveyed. The three known compartments are mutator foci, P granules and Z granules. Together these three compartments can be thought of as a 'germ granule'. Mutator foci are not a 'specific germ granule'.

7. Could the authors please comment what is known about the two classes of CSR-1 targets: 1) CSR-1 promotes transcription of how many germline targets, 2) CSR-1 has been previously shown to promote degradation of mRNAs for how many germline targets, 3) perhaps add a venn diagram showing previously published data for up or down-regulated genes in CSR-1 KO and CSR-1 slicer dead. Is there overlap here? Presumably, some germline genes whose mRNA is sliced and downregulated by CSR-1 may also have its transcription be protected by CSR-1 anti-silencing activity? Then later on, contrast previous results with those presented here.

8. Line 100. It would be helpful if the exact phenotype differences between CSR-1 KO and slicer-dead could be discussed. Are both 100% maternal effect embryonic lethal? Do some slicer-dead homozygotes mothers give rise to embryos that mature to become adults? This will help the reader understand how the authors stage the distinct mutants.

9. 'Therefore we conclude that CSR-1 slices a subset of mRNA targets having abundant 22G RNAs'. Perhaps clarify this by stating 'Therefore, our results support a previously developed model that CSR-1 slices mRNA targets', then point out how your study advances what is already known.

10. 'Thus, CSR-1 slicer activity negatively regulates the expression of it's own interactors'. What about RNA-dependent CSR-1 interactors? Perhaps clearly state that CSR-1 slicer was previously shown to regulate CSR-1 and that subsequent mass spectrometry results allowed you to conclude that CSR-1 interactors were affected.

11. What fraction of CSR-1 targets do not interact with CSR-1? In other words, is the post-transcriptional mRNA degradation function of CSR-1 mostly dedicated to regulating the CSR-1

pathway itself?

12. 'The main role of CSR-1 catalytic activity is to control the accumulation of 22G RNAs'. Please clarify '22G RNAs that associate with CSR-1'?

13. Section entitled 'CSR-1 protects a subset of oogenic targets from piRNA mediated silencing'. Perhaps present this comparison of KO versus slicer first, as it might make it easier for the reader to understand the significance of the slicer section?

14. The HRDE-1 IP experiment provides elegant support for the anti-silencing hypothesis of CSR-1. Please comment if this is the first instance where this has been shown?

15. Line 154 'The global reduction of CSR-1-bound 22G-RNAs observed in CSR-1 mutants' Please indicate if this includes targets that are transcriptionally upregulated and post-transcriptionally down-regulated by CSR-1 slicer activity.

16. What is the phenotype caused by CSR-1 auxin depletion?

17. 'Implying that EGO-1 is exclusively responsible for synthesis of CSR-1 22G RNAs in both WT and *csr-1*'. Instead state 'may be exclusively', as 3' UTRs retain 22G RNAs when EGO-1 RNAi is performed.

18. Why not look at *ego-1* RNAi alone to see if gene body and UTR RNAs are reduced when CSR-1 is wildtype? If *rrf-3* promotes 3' UTR biogenesis of primary CSR-1 siRNAs, why not look at *rrf-3* single and *ego-1 rrf-3* double mutant small RNAs?

19. 'These results highlight that CSR-1 22G RNA biogenesis occurs in cytosol independently of germ granules'. Does P granule RNAi disrupt mutator foci or Z granules? If not, the term 'germ granules' in this sentence is inaccurate.

20. In Fig. 3a, it appears that some perinuclear CSR-1 remains (if you zoom in). Hence, the conclusion that cytosolic CSR-1 is unaffected but germ granule localization is eliminated is uncertain, and this contributes to a major conclusion of this study. If the authors perform RNAi of Z granule or Mutator foci proteins, this might reveal if the remaining perinuclear CSR-1 localized to other foci. Note that there is previously published data for Z foci that may suggest that CSR-1 partially localizes to Z granules and partially to P granules.

21. Line 209 'our data thus far suggests that CSR-1 22G rnas are generated in the cytosol'. 'Might be generated in the cytosol' (see point 20).

21. 'CSR-1 ADH showed reduced purification with ribosomal proteins and increased purification with germ granules and localizes primarily to germ granules (Ex data 5b)'. This important image should be shown in Fig. 4. It appears that there are very large amounts of CSR-1 slicer dead, which may be consistent with the idea that CSR-1 downregulates itself. What are the large CSR-1 granules here? Perhaps show Piwi or PGL-1 IF to determine what the CSR-1 slicer dead foci are?

22. Why does ribosomal protein association suggest a cytoplasmic location? It is known what fraction of ribosomes are associated with germ granules, what fraction are perinuclear and possibly next to germ granules? Mass spectrometry of HRDE-1 has revealed association with several small and large subunit ribosomal proteins, for the nuclear factor HRDE-1. The conclusion that the fraction of CSR-1 associated with ribosomes must be in the cytoplasm rather than in germ granules does not seem the only conclusion one could draw. What if the small fraction of ribosomes that associates with CSR-1 is in a germ granule compartment like Z granules or Mutator foci? What if the small amount of CSR-1 that remains perinuclear upon RNAi of P granules is the fraction that associates ribosomes and promotes 22G RNA production? It is even formally possible that the fraction of CSR-1 that associates

with ribosomes to promote 22G RNA production is so small that it would be difficult or impossible to see by IF. Most ribosomes are thought to be in the cytoplasm, but what if some ribosomes, which are assembled in the nucleolus, associates with CSR-1 in the nucleus prior to promoting 22G RNA production via EGO-1??

23. If P granules are devoid of actively translating mRNAs, might they contain (a low level of) stalled translating mRNAs that are complexes with ribosomes?

24. Discussion 'In this study, we determined the rules governing germline mRNA targeting by CSR-1 and addressed the long-standing paradox of CSR-1 function as anti-silencer or a slicer'. This is an appropriate summary of the duality being investigated that might be helpful to state in the abstract. That said, if slicing regulates both classes of target, then how has the paradox been resolved?

25. 'Synthesis of 22G RNAs occurs in the cytosol on translating mRNA templates, whereas germ granules are dispensable for CSR-1 regulation.' This is implied by phasing, but not supported by an independent line of evidence showing that polysome fractions with stalled ribosomes associate with CSR-1 and EGO-1.

26. 'In this study, we have established that CSR-1 slices targets mRNAs to trigger generation of RDRP-dependent 22G RNAs on the gene body.' Perhaps clarify that a previous study reported that CSR-1 slices some targets and reduces their mRNA levels, whereas this study extends this work by . . .'

27. 'This is consistent with previous rdrp analysis showing that non-polyadenylated 3' oh ends of RNAs served as better substrates for 22G synthesis'. Related to this, it has previously been proposed that CSR-1 may slice histone mRNAs at their 3' ends. Also, it has been reported that CDE-1 adds poly U tails to CSR-1 RNAs. If so, is it possible that 22G RNAs with poly U tails are the primary siRNAs for CSR-1, which can align with the 3' end of mRNA poly(A) tails? Have the authors looked at the residual 22G RNAs present in 3' UTRs upon EGO-1 RNAi to see if they have a signature that suggests how they are created if it is not phasing? It might be helpful to take the 10 most abundant 3' UTR and create a map of the locations of these residual RNAs and their sequences. Also, are there any small RNAs that possess poly-U tails with 22G 5' ends that might be part of the residual 3' UTR small RNA population? Alternatively, could the ribosomal stop codon or the poly(A) addition signal serve as markers that promote biogenesis of primary CSR-1-associated 22G RNAs?

28. 'Therefore, sequences that promote ribosome stalling promote targeting by CSR-1'. Perhaps add more detail here. Ribosome stalling creates a signal that might all for mRNAs with CSR-1 associated on their 3'UTRs to recruit EGO-1 and somehow encourage EGO-1 to initiate 22G synthesis once ribosome stalling has been relieved?

Minor comments:

1. Line 93. 'Biogenesis of 22G-RNAs on the coding sequence'. Do the authors mean 'antisense to the coding sequence'?

2. Line 106. 'At more advanced stages compared to wildtype'. Do the authors mean 'at more advanced ages'?

3. Line 106. 'Increased accumulation of oocytes compared to wt'. Could the authors show representative images of wt vs slicer vs ko?

4. Line 119. 'Csr-1 slicer displayed a global loss of 22G RNAs'. Could the authors specifically state in the text how many CSR-1 targets (transcriptional or post-transcriptional) are unaffected by CSR-1

slicer-dead?

5. What are the 128 22G RNAs with increased levels of 22G RNAs in CSR-1 slicer-dead? Are these bona-fide targets of wildtype CSR-1?
6. What about the rest of the 1536 22G RNAs with a 2-fold reduction in CSR-1 slicer dead that do not have reduced mRNA levels? Do some of these have reduced mRNA levels because some CSR-1 slicer activity promotes the anti-silencing function of CSR-1?
7. It looks like 434 mRNAs are post-transcriptionally regulated by CSR-1 slicer-dead. Why are 119 targets observed in panel B and 434 in C-F?
8. Does panel 1F show increased transcription of targets of very rare 22G RNAs? Should this be added to the concluding sentence on line 134?
9. 'We confirmed that optimal/non-optimal codons correlated with tRNA copy number'. Does tRNA copy number correlate with germline expression of tRNAs?
10. 'In the absence of CSR-1 protein, a subset of CSR-1 targets is misrouted' Which subset?
11. 'Therefore . . . CSR-1 can also license the transcription of germline genes'. Perhaps clarify by stating 'this was previously hypothesized based on transgene analysis and shown directly here'?
12. All all germline genes protected by CSR-1? If not, what about those that are not?
13. Line 354. 'Most germline expressed mRNAs' what fraction?

REVIEWER COMMENTS

Reviewer #1 (Remarks to the Author):

The Argonaute, CSR-1, is quite mysterious, in that it both protects transcripts from piRNA-mediated slicing and also itself slices target mRNAs. In this manuscript, Singh et al design a series of elegant experiments to address this long-standing enigma. First, they demonstrate that the slicing activity of CSR-1 is required to promote 22G-RNA biogenesis, but that only RNAs with very high 22G levels are down-regulated post-transcriptionally by CSR-1. They then further demonstrate that low translation efficiency promotes CSR-1 22G-RNA abundance and that increasing translation efficiency of a CSR-1 target disrupts 22G biogenesis and function. This work will certainly influence the way the field thinks about and understands the role of CSR-1.

We thank the Reviewer #1 for acknowledging the impact of our study in elucidating the long-standing enigma of CSR-1 catalytic function and the biogenesis of CSR-1 22G-RNAs.

Major comments –

Are there distinguishing features for the genes that are protected by CSR-1 from piRNA mediated silencing (Fig 1g-h)? For example, high or low levels of CSR 22G siRNAs? Why are only 227 genes falling into this class when protection from piRNA-mediated silencing seems to be a major theme of previous CSR-1 work?

We have included in Supplementary Fig. 4f the enriched categories for these genes. The genes that appear to be protected by CSR-1 from piRNAs are enriched for oogenic mRNAs (Supplementary Fig. 4f), and they include targets with both high and low levels of CSR-1 22G-RNAs (Supplementary Fig. 4f). Given that oogenic genes initiate their transcription during the developmental timepoint analyzed (44hph), we predict that more oogenic genes might be protected by CSR-1 at later timepoints (48hph). However, we couldn't analyze worms during oocyte production (at 48hph) because of the phenotypic delay observed in CSR-1 mutants (Supplementary Fig. 2c-e). This might be one of the reasons why only few hundreds of genes are detected in our assay. In addition, multiple mechanisms have been proposed to protect germline mRNA from piRNA silencing, including 1) PATCs sequences in introns (Zhang et al., 2018), 2) unknown elements in exonic sequences (Seth et al., 2018), and CSR-1 targeting to mRNAs (Shen et al., 2018). Thus, it is likely that the removal of CSR-1 protein might not be sufficient to unprotect all the germline mRNAs. Lastly, the protective role of CSR-1 from piRNA silencing has been demonstrated on single-copy transgenes, as the Reviewer #1 mentioned, and the protective function of CSR-1 on endogenous germline genes is still largely uncharacterized. We have now included these points in the results and discussion section.

Line 156 – The authors claim that ADH CSR-1 does not show any loss of 22G-RNA binding, however small RNA-seq would not necessarily give a quantitative answer to this question (both because PCR amplification is used for library generation and because analysis generally requires normalizing to library depth).

By immunoprecipitating WT and ADH CSR-1, followed by radiolabeling of co-precipitating small RNAs, the authors could get a more accurate and quantitative assessment of how the ADH mutant affects small RNA binding.

We thank the Reviewer #1 for this comment. In our small RNA sequencing method (sRNA-seq), we remove PCR duplicates by using the unique molecular identifiers (UMI) present in 5' and 3' adaptors. Also, our sRNA-seq method allows the cloning of triphosphate 22G-RNAs as well as miRNAs, piRNAs and other monophosphate short RNAs. These other class of small RNAs, which are not bound by CSR-1, are still present in CSR-1 IPs and are used to normalize among replicates, inputs vs IPs, and mutant conditions. Therefore, we believe that our comparison between CSR-1 WT and ADH IPs is accurate. Our data in fact show that despite the reduction in total 22G-RNAs in *csr-1* catalytic mutant, which primarily occurs on the coding sequence of CSR-1 targets, IPs of CSR-1 ADH show enrichment of 22G-RNAs in IPs compared to the input (Supplementary Fig. 5a), suggesting that CSR-1 ADH is capable of loading 22G-RNAs.

To corroborate the sequencing results with an independent method, we have performed, as suggested by the Reviewer #1, a 5'-P³² radiolabeling of co-precipitating small RNAs with either WT or catalytic mutant CSR-1 protein. In accordance with the sequencing results, these new experiments show that the CSR-1 ADH is not impaired in small RNA binding and it actually appears that the signal is stronger than the one from WT CSR-1 IPs, suggesting that either the loading of small RNAs is more efficient or that the small RNAs loaded by the CSR-

1 ADH are more stable than the ones loaded into the WT protein (Supplementary Fig. 5b). We also observed a statistically significant enhanced binding of 22G-RNAs in catalytic mutant CSR-1 protein by sRNA-seq, when we measure binding efficiency of WT and catalytic mutant CSR-1 by comparing enrichment in IP compared to input (Supplementary Fig. 5a). Given that *csr-1* is one of the upregulated CSR-1 targets and that CSR-1 ADH is highly enriched in P granule compared to CSR-1 WT, this can explain differences in loading and/or stability of 22G-RNAs in CSR-1 ADH compared to CSR-1 WT. We have now included a better clarification in results section and the new experiment in Supplementary Fig. 5b.

In the auxin-mediated CSR-1 depletion (Fig 2d), why are CSR-1 siRNAs still present at the 3' ends of CSR-1 targets. Are these bound by low levels of CSR-1 (but not sufficient to promote cleavage/spreading) or are they bound by another Argonaute protein?

To answer reviewer's point, we have now added a new comparison of small RNA profile in CSR-1 knockout, which lacks CSR-1 protein to further rule out the presence of low levels of CSR-1 in auxin treatment (Fig. 2d-e). We show that the 3'UTR 22G-RNAs are also present in absence of CSR-1 protein and that the biogenesis of these 22G-RNAs exclusively depends on the RdRP EGO-1 (Fig. 2d-f). We have added these new experiments in the result section. Therefore, we excluded that these remaining 22G-RNAs at the 3' end of mRNA targets are loaded by low level of CSR-1 in the auxin-mediated CSR-1 depletion. Whether these EGO-1-dependent 22G-RNAs might be loaded by another Argonaute in absence of CSR-1 is a possibility. In fact, we have observed that in the absence of CSR-1, 22G-RNAs from CSR-1 targets are bound by nuclear argonaute HRDE-1 which acts downstream of piRNA pathway. In WT background, HRDE-1 IP shows binding of only 58 CSR-1 target genes, however, in CSR-1 knockout, HRDE-1 bound 22G-RNAs from 546 CSR-1 targets. We have now included this in results (Supplementary Fig. 4g).

The knockdown of P granule components to demonstrate that CSR-1 22G-RNA biogenesis occurs in the cytoplasm is a beautiful experiment, but I see two caveats that should be discussed. First, some CSR-1 localization is still visible in the perinuclear space, so it remains possible that some 22G-RNA biogenesis is occurring here. Second, biogenesis of CSR class 22G-RNAs in the absence of P granules doesn't preclude the possibility that 22G-RNA biogenesis does occur in P granules when they are present, they are just not necessary.

We thank the reviewer #1 for appreciating our experiments to show that the gene regulatory function of CSR-1 slicer activity and 22G-RNA biogenesis occurs in the cytosol. We agree with the Reviewer #1 that we can still observe some perinuclear localization of CSR-1 in the image shown. This is because we do not obtain 100% knock down during RNAi treatment. However, we have now visualized M, Z, and P granules and show that they are all affected at similar degree together with CSR-1 granule localization (Fig. 3a). Given that the reduction in these granules, even if not 100%, is sufficient to severely deplete piRNA-dependent 22G-RNAs (Fig. 3c), we believe that we should have been able to detect some changes in CSR-1 22G-RNAs accumulation if the biogenesis occurs primarily in P granules. Additionally, there is also no upregulation of CSR-1 mRNA targets (Fig. 3d, S7a). Finally, only CSR-1 22G-RNAs, and not piRNA-dependent 22G-RNAs loaded by HRDE-1, shows 3-nt periodicity typical of ribosome footprint (Fig. 4e, S7d). We now also show that CSR-1 and EGO-1, but not PIWI, are present in polysome fractions (Figure 4d). Therefore, we conclude that the majority of CSR-1 22G-RNAs synthesis and CSR-1 mRNA regulation occurs in the cytosol co-translationally. However, we also agree with the Reviewer #1 and cannot exclude that a minor fraction of CSR-1 22G-RNA synthesis may occurs in P granules, with another mechanism yet to be identified. We have extended the discussion of our experiments in the manuscript to clarify these points raised by the Reviewer #1.

I don't fully follow how Fig 4c-d show that the synthesis of CSR 22G-RNAs is from translating RNAs. Is the proposal that the RdRP is initiating 22G biogenesis just 5' or just 3' of a translating ribosome? Presumably RdRP initiation is limited to starting at a G, which would restrict the possible initiation sites. Is there a physical interaction between the RdRP and the ribosome? Similar question for 5f – how is a 22G siRNA synthesized within the ribosome protected region? Especially if the ribosome and RdRP are moving in opposite directions on the mRNA?

We thank the Reviewer #1 for allowing us to clarify the data presented in Fig. 4c-d and Fig. 5f. In old Fig. 4d (new Fig. 4e) we show that the CSR-1-bound 22G-RNAs have a characteristic three-nucleotide (3-nt) periodicity pattern typical of ribosomal footprints displayed in Ribo-seq reads. In contrast, HRDE-1-bound 22G-RNAs are randomly distributed, indicating that the results obtained with CSR-1 22G-RNAs are not due to some sequence biases. These results suggest that the synthesis of CSR-1 22G-RNAs might occur co-translationally. To strength our conclusion, we have now shown in Fig. 4d, that CSR-1 and EGO-1 are present in the polysome mRNA fractions, suggesting they interact with translating mRNAs. In contrast, PIWI or another P-granule marker PGL-1 are not present in the polysome fraction, which further suggests that the biogenesis of the piRNA-dependent 22G-RNAs is occurring in

the P granules on untranslated mRNAs. A similar 3-nt periodicity has been observed for decapped mRNA that undergo 5'–3' exonucleolytic mRNA decay (Pelechano et al., 2015). Here, the authors observed a bias at the 5'-end of the ribosome suggesting a general 5'–3' mRNA co-translational decay. The results in old Fig. 4c (new Supplementary Fig. 7d) shows distance between the 5' end of 22G-RNAs and RPFs showing a periodic pattern of 3 nt used to calculate periodicity shown in Fig. 4e but does not show any biases along the ribosomal protected fragment. However, in Fig. 5f we show that there is a bias for the 3' end of ribosomal protected fragment in presence of bad codons at A or P site of ribosome, which usually pause the ribosome movement. This result led us to conclude that CSR-1 22G-RNAs might be preferentially initiated at the 3' end of the ribosome and that the pausing of the ribosome might facilitate CSR-1 mRNA cleavage and the initiation of 22G-RNA synthesis by the RdRP EGO-1. We did not observe any specific bias at the last position of RPF (Supplementary Fig. 8f). We still don't know how the small RNAs are synthesized by the RdRP starting from the 3' of the ribosome (whether the ribosome is disassembled etc.). One possibility is that CSR-1 and EGO-1 might coordinate their activity with the ski complex, which mediate the clearance of stalled ribosomes from mRNA in 3'→5' direction and facilitate exosomal degradation (Zinoviev et al., 2020). Additionally, ribosome-phased endonucleolytic cuts possibly produced by the ribosome at the exit site of the mRNA ribosome channel has been recently documented in a process called ribothrypsis (Ibrahim et al., 2018). We included changes in results and discussion section to improve the clarity of our data in our revised version of the manuscript.

Minor comments –

How does this new list of CSR-1 target genes at L4 stage compare to previous CSR-1 target lists (Claycomb, 2009 for adults and Conine 2013 for adult males)?

We have now included an overlap with previous CSR-1 22G-RNA targets in Supplementary Fig. S3a.

Extended Data Fig. 2f is never referenced in the text. Also, for this figure, I'm assuming that grey dots represent all CSR-1 target genes, but this is not explicitly stated in figure or figure legend.

We thank the Reviewer #1 for noticing this mistake. We have corrected it in the revised version of the manuscript. Grey dots are all proteins detected in Mass spec experiment in either CSR-1 IP or control IP, and only enriched proteins in CSR-1 IP with at least 1 log₂ fold change are considered CSR-1 targets, which are marked by the number in parenthesis in the quadrant $x \geq 1, y \geq 1$.

Extended data Fig 6a does not seem like the right figure to demonstrate that 22G-RNA levels are independent of mRNA abundance. Maybe a scatterplot of 22G abundance vs mRNA abundance would be a clearer way to make this point.

We agree with the Reviewer #1 and thanks for this suggestion. We have now included a scatter plot to display this data as suggested by the Reviewer #1 (Supplementary Fig. 8a).

Reviewer #2 (Remarks to the Author):

The authors of this nicely written manuscript address the biogenesis of coding sequence derived antisense 22G-RNAs that are loaded in CSR-1 in the germline of *C. elegans*. This is an important area of investigation because unlike the CSR1 22G-RNAs that target repetitive elements, the biogenesis and function of 22G RNAs that target genic mRNA is a mystery. In this paper, the authors find that the biogenesis of CDS-derived antisense 22G RNAs is cotranslational and triggered by the cleavage activity of CSR1 itself in the vicinity of ribosomes that occupy suboptimal codons (and hence are slower). They show that such biogenesis occurs in the cytosol (as would be expected) and not in germ granules. By further investigating the differences between the molecular phenotypes of catalytically inactive CSR1 and CSR1 KO they identify a catalytic-independent function of CSR1, which is to prevent chromatin silencing by piRNAs. Overall, the study is nicely performed, the data and analyses are of high quality and support the authors' main conclusions.

We thank the Reviewer #2 for the nice summary and for appreciating the high quality of our data and analysis.

Reviewer #3 (Remarks to the Author):

A study by Cecere and colleagues presents a comprehensive analysis of the effects of slicer activity on small RNAs that interact with the CSR-1 Argonaute protein, which suggest that it can act to promote biogenesis of small RNAs that induce mRNA decay when target mRNAs stall during translation. This paper addresses an apparent paradox regarding the multifunctional CSR-1 Argonaute protein, which has been previously shown to possess an ‘anti-silencing’ activity that promotes transcription of germline genes but also has the ability to slice and degrade some germline mRNAs, including those that regulate chromosome segregation during mitosis in embryos. The authors demonstrate that the RDRP EGO-1 promotes siRNA biogenesis in response to CSR-1 slicing of mRNAs, such that small RNAs homologous to the coding sequence mRNAs are depleted when EGO-1 is knocked down when CSR-1 is slicer-dead. The authors provide several lines of evidence that CSR-1-dependent coding sequence RNAs are phased such that they commonly begin at rare codons where ribosomes may transiently stall. They demonstrate that mRNA levels can be tuned by altering codon bias. The authors conclude that ribosomal mRNAs are likely to be targeted by CSR-1 slicer activity for cleavage and small RNA biogenesis in the cytoplasm, in part based on localization of CSR-1 by indirect immunofluorescence. I am uncertain about the strength of the conclusion that cytoplasmic CSR-1 is relevant to reducing mRNA, but many of the conclusions of this study are novel and interesting. The writing of this manuscript could also be improved in other places. I suggest revision of this manuscript by clarifying the background and significance of the data. It may also be helpful to re-frame the concept that the authors provide for CSR-1 primary siRNAs, in part by carefully examining small RNAs that are created at 3’ UTRs.

We thank the Reviewer #3 for appreciating the novelty of our findings and for the careful and insightful review of our manuscript. Thanks to the Reviewer’s constructive comments we have strengthened our conclusion on the cytoplasmic function of CSR-1 in 22G-RNA biogenesis and mRNA regulation and clarified the background and significance of the data following the Reviewer’s rigorous suggestions. Please find below our responses to specific comments.

Major comments:

1. It might be worth citing the original papers by Maine and Sundaram that identified CSR-1 and EGO-1 as part of a complex of proteins that functions together during development, the Maine paper pointing to a role in regulation of the epigenome that may not be understood yet. One possibility is that the effects observed by Maine and Sundaram are due to excess levels of mRNAs targeted by CSR-1 and EGO-1.

We thank the Reviewer #3 for this suggestion. We have now included the work by Maine et al., 2005 in the discussion. We indeed think that the upregulation of CSR-1 targets might indirectly cause some chromatin defects previously documented.

2. Because of the paradoxical nature of CSR-1 functions, it might be worth pointing out that although the original observations of Claycomb and Mello that the CSR-1/EGO-1 complex physically associates with condensed chromosomes, that this association is likely relevant to the anti-silencing role of CSR-1 in promoting transcription. Then, you can clearly explain that slicer activity of CSR-1 has been previously linked to mRNA destruction of genes relevant to mitosis, which is distinct from a role for CSR-1 in transcriptional activation.

We thank the Reviewer #3 for this suggestion. We have now referred to the observation by Claycomb and Mello in the introduction.

3. Line 67. ‘Targeting by CSR-1 22G RNAs can function as an anti-silencing mechanism to protect germline mRNAs from piRNAs silencing . . . Germline mRNAs remain protected from piRNAs silencing even in the absence of CSR-1’. These sentences are confusing based on the paradoxical function of CSR-1 in regulating gene expression. The CSR-1 antisilencing mechanism does not protect germline mRNAs from piRNA silencing, it protects germline expressed genes from transcriptional silencing by piRNAs. As piRNAs promote transcriptional silencing, germline genes remain protected from piRNA silencing in the absence of the CSR-1 anti-silencing activity.

We thank the Reviewer #3 for constructive suggestion. We have now extensively revised the introduction keeping in mind Reviewer’s suggestions and have discussed anti-silencing function of CSR-1 either in the nucleus (Tyc et al. 2017, Akay et al. 2017) or in the P granules (Seth et al., 2018).

4. 'To what extent endogenous germline genes are regulated by the antagonistic functions of CSR-1 and piRNA pathways remains elusive'. Perhaps comment here about the rpm of CSR-1 small RNAs that promote anti-silencing of germline genes: is it known that they are rare?

It was not known whether genes protected by CSR-1 has lower or higher 22G-RNAs. However, we have now discussed in the results that the gene we found to be protected by CSR-1 do not show any specific enrichment for lower or higher antisense 22G-RNA distribution (Supplementary Fig. 4f).

5. 'Worms lacking CSR-1 protein display downregulation of germline CSR-1 targets, those expressing CSR-1 catalytic mutant show upregulation of its germline gene targets.' This is confusing. Does lack of CSR-1 only cause downregulation of germline CSR-1 targets? Does lack of CSR-1 not also result in upregulation of some targets (that are targeted by slicer activity)? It might be helpful to state here how many genes are known to be downregulated or upregulated by CSR-1, which will help the reader understand the significance of your advance.

We agree that the sentence might be confusing, and in fact this sentence in the introduction is referring to the previous work showing either global downregulation of CSR-1 targets (using CSR-1 KO, Claycomb et al., 2009) or global upregulation of CSR-1 targets (using overexpression of CSR-1 ADH in *csr-1* KO, Gerson-Gurwitz et al. 2016). We have now better explained the different studies in the introduction.

In our study, we have observed that severe developmental defects might have contributed to obtain such different gene expression profiles in previous studies. For this reason, in our manuscript we are carefully evaluating the role of CSR-1 KO or ADH (generated by CRISPR- Cas9) in gene regulation by using a sorting strategy to precisely obtain worms at similar developmental stage without developmental abnormalities. By doing this, we have discovered that CSR-1 targets with abundant 22G-RNAs are indeed upregulated in both CSR-1 KO and CSR-1 ADH. However, we have also identified a subclass of genes downregulated in CSR-1 KO but not in CSR-1 ADH (Fig. 1h-i). As requested by the Reviewer #3, we have provided the number of misregulated genes in the text (Supplementary Fig. 2a-b) and the list of these genes in the Supplementary Data 1.

6. 'However, disruption of a specific germ granule, the mutator foci'. The term 'Germ granules' is confusing in this manuscript. *C. elegans* germ granules are composed of at least three compartments that lie next to each other and be distinct but also may physically overlap. This needs to be clearly conveyed. The three known compartments are mutator foci, P granules and Z granules. Together these three compartments can be thought of as a 'germ granule'. Mutator foci are not a 'specific germ granule'.

We agree with the Reviewer #3 and we are now referring to the germ granules only when we want to talk in general of all the three compartments: P, Z, and M (also known as mutator foci) granules. We have clarified in the manuscript this distinction.

7. Could the authors please comment what is known about the two classes of CSR-1 targets: 1) CSR-1 promotes transcription of how many germline targets, 2) CSR-1 has been previously shown to promote degradation of mRNAs for how many germline targets, 3) perhaps add a Venn diagram showing previously published data for up or down-regulated genes in CSR-1 KO and CSR-1 slicer dead. Is there overlap here? Presumably, some germline genes whose mRNA is sliced and downregulated by CSR-1 may also have its transcription be protected by CSR-1 anti-silencing activity? Then later on, contrast previous results with those presented here.

One issue in comparing dataset from different studies is that they have been generated using different strains at different developmental stages. This is why in this study we have measured transcription, mRNAs, translation and small RNAs from KO and ADH mutant generated in our lab at a precise developmental stage. For instance, Cecere et al. 2014 have measured transcription in an hypomorphic strain of CSR-1 at L3 larva stage. In Claycomb et al. 2009, they have done microarray of CSR-1 KO using young adults and in work by Gerson-Gurwitz et al, 2016, they have used a *csr-1* KO strain complemented with a transgenic expression of CSR-1 ADH at young Adult stage. We have observed that either CSR-1 KO or ADH have developmental defects when they reached Young adult stages. For this reason, we have collected our worms at late L4 stages before oogenesis starts. These mutants have in fact strong delay in starting oogenesis and at later time points they actually display even more oocytes than the wild type (Supplementary Fig. 2c-f). We believe that these developmental defects might have skewed many of the previous gene expression studies. Nonetheless, we have included in the result section the analysis of the previous dataset and compared with our data as suggested by the reviewer #3 and included new figures (Supplementary Fig. 2a, b).

8. Line 100. It would be helpful if the exact phenotype differences between CSR-1 KO and slicer-dead could be discussed. Are both 100% maternal effect embryonic lethal? Do some slicer-dead homozygotes mothers give rise

to embryos that mature to become adults? This will help the reader understand how the authors stage the distinct mutants.

We have shown in a recently published study from our lab (Quarato et al., 2021), that even if CSR-1 ADH is slightly more fertile than CSR-1 KO they both display 100% embryonic lethality (Supplementary Fig. 2a, b in Quarato et al., 2021). Therefore, for our experiment we start with heterozygote CSR-1 KO and CSR-1 ADH and we use a sorting strategy that allows us to collect almost pure population of M⁺/Z⁻ mutants. The synchronized larvae will grow and will be sorted again to select only worms perfectly staged at late L4 (44hph). This strategy allowed us to avoid collecting worms with strong developmental defects and developmental delay. We have now added a schematic in Supplementary Fig. 2g explaining our strategy to collect CSR-1 KO and CSR-1 AHD mutants and explained the same in result section.

9. ‘Therefore we conclude that CSR-1 slices a subset of mRNA targets having abundant 22G RNAs’. Perhaps clarify this by stating ‘Therefore, our results support a previously developed model that CSR-1 slices mRNA targets’, then point out how your study advances what is already known.

Correct, we have now included the clarification as suggested by the Reviewer #3.

10. ‘Thus, CSR-1 slicer activity negatively regulates the expression of it’s own interactors’. What about RNA-dependent CSR-1 interactors? Perhaps clearly state that CSR-1 slicer was previously shown to regulate CSR-1 and that subsequent mass spectrometry results allowed you to conclude that CSR-1 interactors were affected.

We have rephrased the sentence as suggested by the Reviewer #3. We focused here only on direct CSR-1 interactors as majority of CSR-1 targets were found to be direct interactors.

11. What fraction of CSR-1 targets do not interact with CSR-1? In other words, is the post-transcriptional mRNA degradation function of CSR-1 mostly dedicated to regulating the CSR-1 pathway itself?

A large number of CSR-1 targets in fact do not interact with CSR-1 (Supplementary Fig. 3c). However, it is important to note that targets are defined by sRNA-seq from CSR-1 IP and interactors from MS/MS analysis of IP which does not have similar dynamic range. Nonetheless, CSR-1 direct interactors are enriched among CSR-1 targets (Fig. 1g). We have discussed this in more details in the result and discussion section.

12. ‘The main role of CSR-1 catalytic activity is to control the accumulation of 22G RNAs’. Please clarify ‘22G RNAs that associate with CSR-1’?

We thank the Reviewer #3 for this observation. We have changed the sentence and specified that those are CSR-1-interacting 22G-RNAs.

13. Section entitled ‘CSR-1 protects a subset of oogenic targets from piRNA mediated silencing’. Perhaps present this comparison of KO versus slicer first, as it might make it easier for the reader to understand the significance of the slicer section?

We thank the Reviewer #3 for this suggestion. Following all the comments from the reviewers, we have significantly revised the introduction and results and further included a dedicated paragraph in discussion highlighting the slicer and anti-silencing role of CSR-1 and we hope the reviewer will find these modifications satisfactory.

14. The HRDE-1 IP experiment provides elegant support for the anti-silencing hypothesis of CSR-1. Please comment if this is the first instance where this has been shown?

We thank the Reviewer #3 for appreciating our result. This is indeed the first evidence of endogenous genes targeted by HRDE-1 in absence of CSR-1, supporting its anti-silencing role. We have specified it in the text.

15. Line 154 ‘The global reduction of CSR-1-bound 22G-RNAs observed in CSR-1 mutants’ Please indicate if this includes targets that are transcriptionally upregulated and post-transcriptionally down-regulated by CSR-1 slicer activity.

The reduction of CSR-1-bound 22G-RNAs includes post-transcriptionally regulated genes by CSR-1 slicer activity and genes where we observed CSR-1 anti-silencing activity, we have included this in Supplementary Fig. 4i.

16. What is the phenotype caused by CSR-1 auxin depletion?

The phenotype caused by CSR-1 auxin depletion has been shown in Quarato et al., 2021 (Fig. 1b, c). If the auxin treatment is initiated at the L1 stage (as shown in this manuscript) there is a reduction in the broodsize and 100% embryonic lethality. Treatment initiated at the L4 stage do not affect the brood size, yet it shows 100 % embryonic lethality. We now have explained this in the manuscript.

17. 'Implying that EGO-1 is exclusively responsible for synthesis of CSR-1 22G RNAs in both WT and *csr-1*'. Instead state 'may be exclusively', as 3' UTRs retain 22G RNAs when EGO-1 RNAi is performed.

We suspect that the retained 22G-RNAs on 3'UTR when *ego-1* RNAi was performed, was due to lack of 100 % penetrance of RNAi. To confirm this, we have now generated an *ego-1* mutant by CRISPR-cas9 and performed sRNA-seq using the EGO-1 KO. Our new results show complete depletion of 22G-RNAs on CSR-1 targets on the coding sequence as well as 3'UTR (Fig. 2d).

18. Why not look at *ego-1* RNAi alone to see if gene body and UTR RNAs are reduced when CSR-1 is wildtype? If *rrf-3* promotes 3' UTR biogenesis of primary CSR-1 siRNAs, why not look at *rrf-3* single and *ego-1 rrf-3* double mutant small RNAs?

Following the Reviewer #3 suggestions, we are now showing 22G-RNA profiles in *ego-1* RNAi in WT, *ego-1* KO, *rrf-3* KO, and *rrf-3* KO with *ego-1* RNAi. These results, show that *rrf-3*, which is responsible for synthesis of 26G-RNAs (Supplementary Fig. 6c), is not involved in the biogenesis of 22G-RNAs from CSR-1 targets and is not contributing to the signal observed in *ego-1* RNAi at the 3'end of the genes (Fig. 2g). These 3'end signal is most likely present because of the not complete depletion of EGO-1 protein. In fact, *ego-1* KO shows complete depletion of CSR-1 22G-RNAs (Fig. 2d).

19. 'These results highlight that CSR-1 22G RNA biogenesis occurs in cytosol independently of germ granules'. Does P granule RNAi disrupt mutator foci or Z granules? If not, the term 'germ granules' in this sentence is inaccurate.

We thank the Reviewer #3 for pointing this out. We have now provided new experiments showing that the quadruple P granule RNAi treatment reduce the P, M, and Z granules at similar levels (Fig. 3a).

20. In Fig. 3a, it appears that some perinuclear CSR-1 remains (if you zoom in). Hence, the conclusion that cytosolic CSR-1 is unaffected but germ granule localization is eliminated is uncertain, and this contributes to a major conclusion of this study. If the authors perform RNAi of Z granule or Mutator foci proteins, this might reveal if the remaining perinuclear CSR-1 localized to other foci. Note that there is previously published data for Z foci that may suggest that CSR-1 partially localizes to Z granules and partially to P granules.

We thank the Reviewer #3 for this observation. We acknowledge that it is in fact important to characterize the effect of the P granule RNAi treatment on germ granules, including Z and M granules. Therefore, we now show that the P granule RNAi treatment destabilize the P, M, and Z granules at similar level (Fig. 3a). Since RNAi is not 100 % effective, we show that the remaining CSR-1 protein colocalize with the remaining P granule protein DEPS-1 (Fig. 3b) upon RNAi treatment. We are also showing that *znfx-1*(Z granule) or *mut-16* (M granule) mutants do not affect CSR-1 localization to P granule (Supplementary Fig. 7b). Based on these new results, we believe that the P granule RNAi treatment reduce the level of CSR-1 localization in P granules. Nonetheless, there is no effect on CSR-1 22G-RNA biogenesis, nor on the mRNA levels of the corresponding CSR-1 targets (Fig. 3 c-d). Conversely, the same treatment is causing a severe downregulation of piRNA-dependent 22G-RNAs (Fig. 3c).

21. Line 209 'our data thus far suggests that CSR-1 22G rnas are generated in the cytosol'. 'Might be generated in the cytosol' (see point 20).

We agree that the P granule RNAi experiment cannot exclude that a biogenesis of CSR-1-bound 22G-RNAs can still occur in P granule. However, the cytosolic biogenesis might contribute to the majority of CSR-1-bound 22G-RNAs. We have modified the text according to Reviewer's suggestion.

21. 'CSR-1 ADH showed reduced purification with ribosomal proteins and increased purification with germ granules and localizes primarily to germ granules (Ex data 5b)'. This important image should be shown in Fig. 4.

It appears that there are very large amounts of CSR-1 slicer dead, which may be consistent with the idea that CSR-1 downregulates itself. What are the large CSR-1 granules here? Perhaps show Piwi or PGL-1 IF to determine what the CSR-1 slicer dead foci are?

We agree with the Reviewer #3 about the importance of old Supplementary Fig. 5b. We have now moved this in Fig. 4 and have included images that show the colocalization of CSR-1 enlarged granules with the P granule protein GLH-1. Given that CSR-1 post-transcriptionally regulate CSR-1 itself and other germ granules components, the catalytic activity of CSR-1 might also be required to maintain germ granules homeostasis. In fact, we believe that these enlarged granules in CSR-1 ADH mutant are not functional like the wild-type granules, and we are characterizing this phenotype in another ongoing study.

22. Why does ribosomal protein association suggest a cytoplasmic location?

We agree that the direct interaction between CSR-1 and ribosomal proteins is not an indication that this interaction is happening in the cytosol on translating mRNAs. For this reason, we have now performed polysome fractionations and observe that CSR-1 and EGO-1 co-purify with translating mRNAs in polysome fractions. In contrast, PIWI or P-granule protein PGL-1 is not enriched in polysome fraction (Fig. 4d).

It is known what fraction of ribosomes are associated with germ granules, what fraction are perinuclear and possibly next to germ granules?

It is currently not known what fraction of ribosomal proteins associate with germ granules. However, in the IPs of ADH, which is predominantly enriched in enlarged granules, most of the direct CSR-1 interacting ribosomal proteins are depleted compared to IPs of WT CSR-1 (Fig. 4b). This experiment suggests that the interaction between CSR-1 and ribosomal proteins might occur in the cytosol.

Mass spectrometry of HRDE-1 has revealed association with several small and large subunit ribosomal proteins, for the nuclear factor HRDE-1.

It is true that often ribosomal proteins can be found in IPs experiments of RNA binding proteins. To strength our conclusion on the direct interaction between CSR-1 and ribosomal proteins we are showing mass spec data of PIWI in RNase vs no RNase condition. As shown in Supplementary Fig. 7c most of the ribosomal proteins interacting with PIWI are depleted upon RNase treatment but not for CSR-1, suggesting that PIWI is not directly interacting with ribosomal proteins. This result, together with the fact that PIWI is not enriched in polysome fractions constitute a good control to strength the conclusion that CSR-1 and EGO-1 directly interacts with ribosomes on translating mRNAs.

The conclusion that the fraction of CSR-1 associated with ribosomes must be in the cytoplasm rather than in germ granules does not seem the only conclusion one could draw. What if the small fraction of ribosomes that associates with CSR-1 is in a germ granule compartment like Z granules or Mutator foci? What if the small amount of CSR-1 that remains perinuclear upon RNAi of P granules is the fraction that associates ribosomes and promotes 22G RNA production? It is even formally possible that the fraction of CSR-1 that associates with ribosomes to promote 22G RNA production is so small that it would be difficult or impossible to see by IF. Most ribosomes are thought to be in the cytoplasm, but what if some ribosomes, which are assembled in the nucleolus, associates with CSR-1 in the nucleus prior to promoting 22G RNA production via EGO-1??

Several lines of evidence suggest that the interaction of CSR-1 with ribosome occurs in the cytosol and not in germ granules: i) only CSR-1 and not another P granule protein, such as PIWI, directly interacts with ribosomal proteins (Supplementary Fig. 7c). ii) CSR-1, but not PIWI or PGL-1, co-purify with polysome mRNAs (Fig. 4d). iii) The interaction between CSR-1 and ribosomal proteins is depleted in CSR-1 ADH IPs and the interaction with germ granule proteins are enriched (Fig. 4b). This correlates with imaging experiments showing that CSR-1 ADH is highly enriched in enlarged P granules (Fig. 4c). Therefore, it is unlikely that the interaction with ribosomal proteins occurs in P granules. We have added these comments in the discussion section. In addition, even if a low level of translation might exist in P granules this has yet to be documented (see the reply to the comment 23).

23. If P granules are devoid of actively translating mRNAs, might they contain (a low level of) stalled translating mRNAs that are complexes with ribosomes?

To our knowledge ribosomal stalling has not been documented in P granule and even though some ribosomal proteins might also localize to P granules they are not in complex within a ribosome on translating mRNAs. Recent

work by the Seydoux lab has shown that P granules contains mRNAs not engaged in translation, and translational activation correlates with P granule exit (Lee et al., 2020). In addition, biochemical and proteomic characterizations of other cytoplasmic granules such as P bodies also show that those granules are depleted of ribosomal proteins (Hubstenberger et al., 2017). We have now included these comments in the discussion.

24. Discussion ‘In this study, we determined the rules governing germline mRNA targeting by CSR-1 and addressed the long-standing paradox of CSR-1 function as anti-silencer or a slicer’. This is an appropriate summary of the duality being investigated that might be helpful to state in the abstract. That said, if slicing regulates both classes of target, then how has the paradox been resolved?

We thank the Reviewer #3 for this suggestion. The two class of genes, the one regulated by slicer activity and the one regulated by anti-silencing are distinct (Supplementary Fig. 4h). We have also included more explanation in discussion section. We still prefer to have this sentence in the summary of the discussion, and leave the abstract more focused on the CSR-1 22G-RNA biogenesis.

25. ‘Synthesis of 22G RNAs occurs in the cytosol on translating mRNA templates, whereas germ granules are dispensable for CSR-1 regulation.’ This is implied by phasing, but not supported by an independent line of evidence showing that polysome fractions with stalled ribosomes associate with CSR-1 and EGO-1.

To strength our finding, we have now included the polysome fraction showing the association of CSR-1 and EGO-1 with polysome, as the Reviewer #3 suggested (Fig. 4d).

26. ‘In this study, we have established that CSR-1 slices targets mRNAs to trigger generation of RDRP-dependent 22G RNAs on the gene body.’ Perhaps clarify that a previous study reported that CSR-1 slices some targets and reduces their mRNA levels, whereas this study extends this work by . . .’

We clarified this as the Reviewer #3 suggested.

27. ‘This is consistent with previous rdrp analysis showing that non-polyadenylated 3’ oh ends of RNAs served as better substrates for 22G synthesis’. Related to this, it has previously been proposed that CSR-1 may slice histone mRNAs at their 3’ ends. Also, it has been reported that CDE-1 adds poly U tails to CSR-1 RNAs. If so, is it possible that 22G RNAs with poly U tails are the primary siRNAs for CSR-1, which can align with the 3’ end of mRNA poly(A) tails?

The poly U tails on CSR-1 22G-RNAs is present at their 3’end and not at their 5’ end, therefore they cannot align on the poly(A) tails. We have now included a graphic representation to help the reader understanding the directionality of our model (Fig. 5f).

Have the authors looked at the residual 22G RNAs present in 3’ UTRs upon EGO-1 RNAi to see if they have a signature that suggests how they are created if it is not phasing?
It might be helpful to take the 10 most abundant 3’ UTR and create a map of the locations of these residual RNAs and their sequences.

We now show that even the 22G-RNAs present in 3’UTR depends on EGO-1 (by profiling 22G-RNAs in *ego-1* KO, Fig. 2d) and the residual 22G-RNAs on *ego-1* RNAi are lost in *ego-1* KO. Nonetheless, we have analyzed the sequence composition of the 3’UTR EGO-1-dependent 22G-RNAs as suggested by the Reviewer #3, and we have not found any sequence biases among them. We have included this in results section (Supplementary Fig. 6d-e).

Also, are there any small RNAs that possess poly-U tails with 22G 5’ ends that might be part of the residual 3’ UTR small RNA population?

We have now exclusively analyzed the 22G-RNAs containing poly U tails in our dataset and we do not find any particular bias in the one mapping to the 3’UTR compared to the one mapping on the coding sequence. We have included this in results section (Supplementary Fig. 6f-h).

Alternatively, could the ribosomal stop codon or the poly(A) addition signal serve as markers that promote biogenesis of primary CSR-1-associated 22G RNAs?

At the current stage we don’t know what trigger the biogenesis of primary EGO-dependent 22G-RNAs on the 3’UTR. It might be possible that the poly (A) might serve as signal or that poly(A) binding proteins or even other

3'UTR RNA binding proteins might serve to recruit EGO-1 on target mRNAs. The identity of these signals or factors might be revealed in future studies. We have discussed all these possibilities in the discussion section.

28. 'Therefore, sequences that promote ribosome stalling promote targeting by CSR-1'. Perhaps add more detail here. Ribosome stalling creates a signal that might all for mRNAs with CSR-1 associated on their 3'UTRs to recruit EGO-1 and somehow encourage EGO-1 to initiate 22G synthesis once ribosome stalling has been relieved?

We thank the reviewer for this suggestion. We have expanded our discussion and included this possibility.

Minor comments:

1. Line 93. 'Biogenesis of 22G-RNAs on the coding sequence'. Do the authors mean 'antisense to the coding sequence'?

Yes, that's correct we have changed it.

2. Line 106. 'At more advanced stages compared to wildtype'. Do the authors mean 'at more advanced ages'?

We thank the Reviewer #3 for pointing this out. We have now changed it.

3. Line 106. 'Increased accumulation of oocytes compared to wt'. Could the authors show representative images of wt vs slicer vs ko?

We have now included a representative image as suggested by the Reviewer #3 (Supplementary Fig. 2c).

4. Line 119. 'Csr-1 slicer displayed a global loss of 22G RNAs'. Could the authors specifically state in the text how many CSR-1 targets (transcriptional or post-transcriptional) are unaffected by CSR-1 slicer-dead?

We have now clearly included the numbers of targets showing significant loss of 22G-RNAs. Both CSR-1 protected, and sliced targets show loss of 22G-RNAs (Supplementary Fig. 4i)

5. What are the 128 22G RNAs with increased levels of 22G RNAs in CSR-1 slicer-dead? Are these bona-fide targets of wildtype CSR-1?

Yes, those are CSR-1 22G-RNAs antisense to spermatogenic genes. These class of genes are also transcriptional upregulated (see below answer to comment 8). The regulation of spermatogenic transcription by multiple small RNA pathways is currently an ongoing study in our laboratory.

6. What about the rest of the 1536 22G RNAs with a 2-fold reduction in CSR-1 slicer dead that do not have reduced mRNA levels? Do some of these have reduced mRNA levels because some CSR-1 slicer activity promotes the anti-silencing function of CSR-1?

The 1536 CSR-1 22G-RNA targets (with > 2-fold reduction), 119 are 2-fold upregulated and only 1 is 2-fold downregulated in CSR-1 catalytic mutant.

7. It looks like 434 mRNAs are post-transcriptionally regulated by CSR-1 slicer-dead. Why are 119 targets observed in panel B and 434 in C-F?

The 119 targets in panel B are genes upregulated > 2-fold and that have reduced CSR-1 22G-RNAs > 2-fold. The 434 target genes shown in C-F are genes with high abundance of 22G-RNAs (> 150 RPM). In Fig. 1c-f, we have binned CSR-1 targets based on abundance of 22G-RNAs bound by CSR-1 and we see the effect of CSR-1 mutation on the targets in these three categories and we observe that the upregulation is dependent of 22G-RNA abundance as also observed in a previous study (Gerson-Gurwitz et.al, 2016).

8. Does panel 1F show increased transcription of targets of very rare 22G RNAs? Should this be added to the concluding sentence on line 134?

We have currently a manuscript in preparation on these specific rare targets that are transcriptionally upregulated. These are in fact regulated in early timepoints and belong to the spermatogenic expressed genes. We have explained this in the manuscript.

9. 'We confirmed that optimal/non-optimal codons correlated with tRNA copy number'. Does tRNA copy number correlate with germline expression of tRNAs?

We don't have transcriptome of tRNAs on dissected gonads, but we have correlated with GRO-seq data from whole worms, and we show a correlation between copy number and transcription of tRNAs (Supplementary Fig. 8d-e).

10. 'In the absence of CSR-1 protein, a subset of CSR-1 targets is misrouted' Which subset?

We have now analyzed the category of these misrouted targets in HRDE-1, and they are enriched for oogenic mRNAs (Supplementary Fig. 4f). We have specified this in the text.

11. 'Therefore . . . CSR-1 can also license the transcription of germline genes'. Perhaps clarify by stating 'this was previously hypothesized based on transgene analysis and shown directly here'?

We have clarified this statement.

12. Are all germline genes protected by CSR-1? If not, what about those that are not?

There are at least three mechanisms that have been proposed to protect germline mRNAs: 1) PATCs in introns, 2) unknown elements in exons, 3) CSR-1 targeting. We have specified this now in the text.

13. Line 354. 'Most germline expressed mRNAs' what fraction?

We have now included a figure showing CSR-1 targets as a fraction of germline enriched and germline expressed genes (Supplementary Fig. 3b).

REVIEWERS' COMMENTS

Reviewer #1 (Remarks to the Author):

The authors have done a nice job addressing the reviewers' comments. I have a few very minor comments, but otherwise I am satisfied and believe that the manuscript is now ready for publication in Nature Communications.

Line 196 – typo. Should be “occurred antisense to the coding sequence”

Line 248 - It is surprising that MUT-16 foci are reduced in the P granule RNAi experiment, as it has been shown previously that RNAi of P granule components *glh-1* and *pgl-1* does not compromise Mutator foci (Phillips et al, Genes and Dev 2012). The authors may want to comment on this discrepancy. Also, it would be more consistent with the previous literature to refer to the Mutator foci as Mutator foci, rather than M granules.

Line 261 – typo, missing closed parentheses -> (RNA-seq data from40

Reviewer #3 (Remarks to the Author):

Cecere revision:

Cecere and colleagues have done a thorough job of revising their manuscript. The newly expanded introduction explains the background for CSR-1 much better. The authors now provide a convincing and thorough understanding of at least two CSR-1 activities: one that protects germline gene transcription and a second that slices and destabilizes ribosome-associated CSR-1 targets. The authors do a nice job of proving their hypothesis that non-optimal ribosome codons promote CSR-1 slicer activity by codon optimizing the target *klp-7* RNA and then carefully measuring translation efficiency and a marked decrease in synthesis of 22G-RNAs associated with CSR-1, including heterozygous and homozygous lines of the codon optimized target. This is truly a lovely set of results that provides satisfying evidence that is appropriate for Nature Communications or even for Nature for that matter.

Comments:

1. Abstract: “in phase with ribosome translation in the cytoplasm, in contrast to other 22g RNAs mostly synthesized in germ granules”.

It can be formally argued that both germ granules and ribosomes are in the cytoplasm? An alternative might be “in phase with translating ribosomes, in contrast to other 22g RNAs mostly synthesized in germ granules”.

2. ‘Whether CSR-1 22g RNAs are synthesized in germ granules on a specific type of RNA substrates remains to be elucidated’. This might be too specific. Perhaps ‘The subcellular location and RNA substrate used to create 22G RNAs is unknown’.

3. ‘Especially occupying bad codons’. Perhaps especially occupying codons that are difficult to translate’?

4. Can the authors speculate how analysis of protection germline genes by CSR-1 could be studied? Perhaps by eliminating nuclear CSR-1 localization? Or is there another Argonaute domain that might be relevant?

REVIEWERS' COMMENTS

Reviewer #1 (Remarks to the Author):

The authors have done a nice job addressing the reviewers' comments. I have a few very minor comments, but otherwise I am satisfied and believe that the manuscript is now ready for publication in Nature Communications.

We thank the reviewer for appreciating the revised manuscript. We have now made suitable changes for the minor comments of the reviewer.

Line 196 – typo. Should be “occurred antisense to the coding sequence”

Thanks for noticing the mistake, we have now corrected the same

Line 248 - It is surprising that MUT-16 foci are reduced in the P granule RNAi experiment, as it has been shown previously that RNAi of P granule components *glh-1* and *pgl-1* does not compromise Mutator foci (Phillips et al, Genes and Dev 2012). The authors may want to comment on this discrepancy. Also, it would be more consistent with the previous literature to refer to the Mutator foci as Mutator foci, rather than M granules.

In the article by Phillips et al, Genes and Dev 2012, authors have used RNAi against either single or two components of P granule which did not disrupt MUT-16 foci. In the current study, we have used RNAi against four components of P granules, thus seeing a more drastic reduction of MUT-16 foci. We have now included a statement clarifying the same on Page 11.

As suggested by the reviewer, we have changed M granule to Mutator foci.

Line 261 – typo, missing closed parentheses -> (RNA-seq data from40

Thanks for the correction, we have corrected the same.

Reviewer #3 (Remarks to the Author):

Cecere revision:

Cecere and colleagues have done a thorough job of revising their manuscript. The newly expanded introduction explains the background for CSR-1 much better. The authors now provide a convincing and thorough understanding of at least two CSR-1 activities: one that protects germline gene transcription and a second that slices and destabilizes ribosome-associated CSR-1 targets. The authors do a nice job of proving their hypothesis that non-optimal ribosome codons promote CSR-1 slicer activity by codon optimizing the target *kfp-7* RNA and then carefully measuring translation efficiency and a marked decrease in synthesis of 22G-RNAs associated with CSR-1, including heterozygous and homozygous lines of the

codon optimized target. This is truly a lovely set of results that provides satisfying evidence that is appropriate for Nature Communications or even for Nature for that matter.

We are glad that we have been able to satisfactorily address all the comments from the Reviewer 3. We appreciate and thank the reviewer for the positive feedback the reviewer provided us to enable us to improve our manuscript.

Comments:

1. Abstract: “in phase with ribosome translation in the cytoplasm, in contrast to other 22g RNAs mostly synthesized in germ granules”.
It can be formally argued that both germ granules and ribosomes are in the cytoplasm? An alternative might be “in phase with translating ribosomes, in contrast to other 22g RNAs mostly synthesized in germ granules”.

We agree with the reviewer and have included the correction in abstract.

2. ‘Whether CSR-1 22g RNAs are synthesized in germ granules on a specific type of RNA substrates remains to be elucidates’. This might be too specific. Perhaps ‘The subcellular location and RNA substrate used to create 22G RNAs is unknown’.

We have made the change in the manuscript

3. ‘Especially occupying bad codons’. Perhaps especially occupying codons that are difficult to translate’?

We have made the change as per the reviewer’s suggestion.

4. Can the authors speculate how analysis of protection germline genes by CSR-1 could be studied? Perhaps by eliminating nuclear CSR-1 localization? Or is there another Argonaute domain that might be relevant?

We thank the reviewer for this suggestion. We have now discussed this in the Discussion section.